# A modified fractional short circuit current MPPT and multicellular converter for improving power quality and efficiency in PV chain

**Geoffroy Byanpambé**[1]*, **Philippe Djondiné**[2,3], **Golam Guidkaya**[2], **Mohammed F. Elnaggar**[4,5], **Alexis Paldou Yaya**[1,6], **Emmanuel Tchindebé**[1], **Kitmo**[7], **Noel Djongyang**[7]

**1** Faculty of Science, Department of Physics, University of Maroua, Maroua, Cameroon, **2** Faculty of Science, Department of Physics, University of Ngaoundere, Ngaoundere, Cameroon, **3** Department of Physics, Higher Teacher Training College, University of Bertoua, Bertoua, Cameroon, **4** Department of Electrical Engineering, College of Engineering, Prince Sattam Bin Abdulaziz University, Al-Kharj, Saudi Arabia, **5** Faculty of Engineering, Department of Electrical Power and Machines Engineering, Helwan University, Helwan, Egypt, **6** Department of Materials Engineering and Natural Resources Valorization, Laboratory of Applied Physics and Engineering, Avanced School of Mines Processing and Energy Resources, University of Bertoua, Bertoua, Cameroon, **7** Department of Renewable Energy, National Advanced School of Engineering, University of Maroua, Maroua, Cameroon

* geoffroybyanpambe@gmail.com

**Data Availability Statement:** All relevant data are within the manuscript and its Supporting Information files.

## Abstract

This article presents the contribution of multicellular converters in improving of the quality of power produced in photovoltaic chain, with the aim of exploiting the maximum power produced by the photovoltaic generator with low oscillations around of the maximum power point (MPP) at steady state and to reduce switching losses. After modeling the multicellular parallel boost converter, fractional short circuit current (FSCC) MPPT was modified to get an estimated photocurrent as a reference to control the inductance current for good functioning of the converter in pursuit of the maximum power point. To verify the performance of the proposed solution, the system was submitted to irradiance and temperature variations. The simulations carried out in the Matlab/Simulink environment presented satisfactory results of the proposed solution, in comparison with the high-gain quadratic boost converter we have a response time of 0.04 s, power oscillations at maximum point around 0.05 W and efficiency of 99.08%; in comparison with the interleaved high-gain boost converter the results show a response time of 0.1 s for the transferred power, a very low output voltage ripples of 0.001% and 98.37% as efficiency of the chain. The proposed solution can be connected to a grid with a reduction of level of the inverter and active filter.

## 1. Introduction

Electrical energy is an essential element for development and improvement of living conditions of a society; its shortage disrupts household life and leads to a slowdown in activities in certain areas of country. The promotion of the use of renewable energies is among the

**Funding:** This project is supported via funding from Prince Sattam Bin Abdulaziz University, project number (PSAU/2024/R/1445).

**Competing interests:** The authors have declared that no competing interests exist.

initiatives undertaken by several countries to alleviate electrical energy problems while preserving the environment. Several research projects are being carried out to develop the field of renewable energies. Among renewable energy sources, photovoltaic energy presents itself as the most promising starting from its raw material. The design of optimized photovoltaic systems is by nature difficult due to the fact: on the source side, the power produced by a photovoltaic generator strongly depends on environmental conditions (irradiance and temperature) [1–4] but also the overall state (age) of the system; on the load side, which the nature can be continuous or alternative, each has its own behavior which can be random[1,2,5,6]. For optimal operation of the photovoltaic generator (GPV), the introduction of an optimal power point tracker is necessary. The power point tracker built around a static converter acting as a source-load adapter with the aim of forcing the generator to operate at its maximum power point (MPP). For the control of the static converter, several maximum point tracking techniques forcing the photovoltaic generator to follow the optimal point despite the environmental conditions have been developed. [7–12]. However, the maximum power point varies depending on the surface of the photovoltaic generator, the temperature and the irradiance [13–16]; the aim of the control technique is to automatically modify the duty cycle to bring the generator to its maximum operating point whatever the weather conditions or load variations that may occur as illustrated on Fig 1 [17]. Several maximum power point tracking algorithms have been developed [18–44], according classification made by [29] the different technics can be classified in three large groups: Indirect techniques (fraction of $V_{OC}$ and fraction of $I_{SC}$...), direct techniques (perturb and observe (P&O), incremental conductance...) and techniques by intelligent control (fuzzy logic, sliding mode, etc.). Good dynamic behavior is very useful in the event of rapid variation of the source (irradiance) or the characteristics of the load.

The static converter playing the role of interface between the source and the load most used in photovoltaic applications is the boost converter, the basic boost converters although they raise up the voltage at their output, are more suitable for low power applications [45–49]. For high power applications, several converters structures have been used: fly-back converters, buck-boost converters, multi-cell converters, high-gain quadratic boost converters.

Fly-back converters [49–52] first perform a DC-AC conversion to obtain a high amplitude alternative voltage, then an AC-DC conversion; buck-boost converters [53,54] presented high dynamic losses and low efficiency. Multicellular converters appeared in the 1990s, offering the possibility of reducing voltage or current stresses in power switches. Parallelizing the base converters allows the use of low current levels per block to promote long component life and

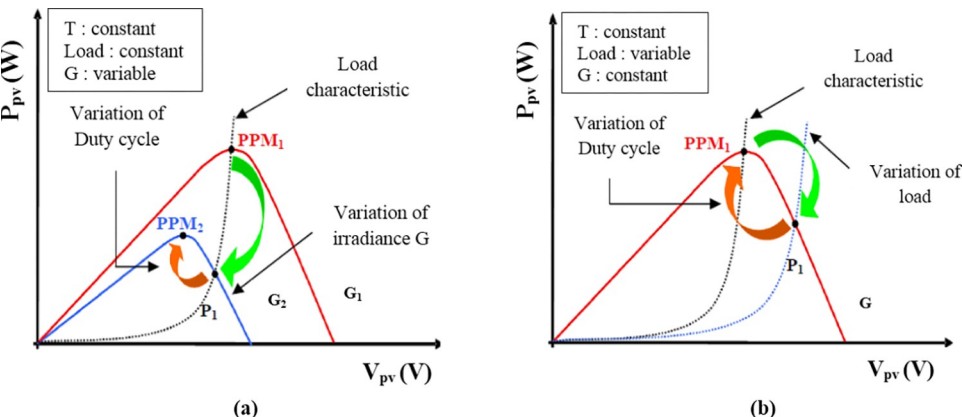

**Fig 1. Research and recovery of the Maximum Power Point [17].** (a)Variation of irradiance. (b) Load variation.

increase the reliability of the base converters [48,55]. Still contributing in high power applications, a single switch quadratic power converter based on a switched capacitor to achieve high voltage gains and low voltage constraints on power components and reduce the complexity of controller designs has presented by [43,49] proposes a high-gain quadratic boost converter (HG-QBC) to overcome the limitations of conventional boost converter and [44] presents a solar-powered interleaved high-gain boost converter (IHGBC) that increases the voltage gain with less ripples in the output voltage compared to existing DC-DC converters.

The massive use of non-linear loads designed using power electronics converters degrades the quality of electrical energy by generating harmonic currents; these power converters can also exhibit chaotic effects or behavior at certain switching frequencies [56,57]. According to [58], increasing the number of switching cells connected in parallel would lead to a reduction in output current ripples (Fig 2). Non-perfect coupling between the photovoltaic generator and the load, oscillations around the maximum point in steady state and occasional loss of tracking the maximum point during rapid change in climatic conditions, and chaotic behavior in static converters are all among the problems which degrade the energy efficiency of a photovoltaic chain.

In this article, the objective is to contribute for improving of energy efficiency and power quality of photovoltaic system by optimizing the power produced based on the operating of GPV on principle of current controlled voltage source (CCVS) and the use of multicellular converters; firstly to improve the indirect short-circuit current fraction (FSCC) method by estimating a photo-current (reference current) to control the inductance current in order to follow the maximum power point (MPP), secondly use the parallel multicellular converter integrating synchronous rectification to reduce switching losses in the switches (by reducing

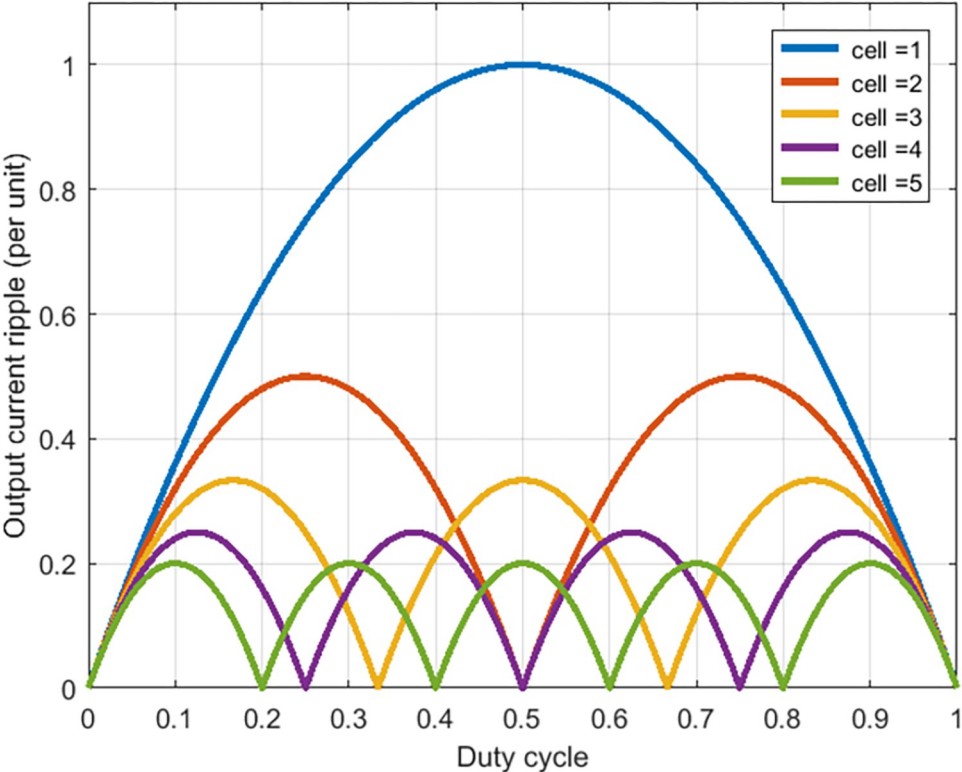

**Fig 2. Output current ripples function of the cells number.**

the switching frequency), operating in interleaving mode for the reduction of power oscillations in steady state.

This paper is structured as follow: section 1 presents the introduction where the state of art is presented, section 2 reserved for models, control techniques and calculation of efficiency, in section 3 is presented the simulation results and discussions, finally conclusion in section 4.

qq2. Method

## 2.1 Photovoltaic generator model

In its constitution, photovoltaic generator (GPV) contains a set of elementary photovoltaic cells; to obtain the desired electrical characteristics (short-circuit current $I_{SC}$, open circuit voltage $V_{OC}$), the elementary cells are connected in series and/or parallel. A photovoltaic cell can be illustrated by its equivalent following diagram [59,60]:

By applying KCL (Kirchhoff's current law) on node N in Fig 3:

$$I = I_{ph} - I_D - I_R \tag{1}$$

with:

$$I_D = I_0 \left( exp \left[ q \left( \frac{V + R_s I}{nKT} \right) \right] - 1 \right) \tag{2}$$

$$I_R = \frac{V + R_S I}{R_P} \tag{3}$$

By replacing $I_D$ et $I_R$ by their expressions in Eq (1) we obtain:

$$I = I_{ph} - I_0 \left( exp \left[ q \left( \frac{V + R_s I}{nKT} \right) \right] - 1 \right) - \frac{V + R_s I}{R_P} \tag{4}$$

Where $I$ is the cell current (A), $I_{ph}$ is the photocurrent (A), $V$ is the cell voltage (V), $R_S$ is series resistance of the cell ($\Omega$), $R_P$ is parallel resistance of the cell ($\Omega$), $T$ is the cell temperature (K), $q$ is electron charge (q = $1,6.10^{-19}$C), $I_O$ the saturation current (A), $K$ is Boltzmann constant (k = $1,3854.10^{-23}$J/K), $n$ is the diode quality factor.

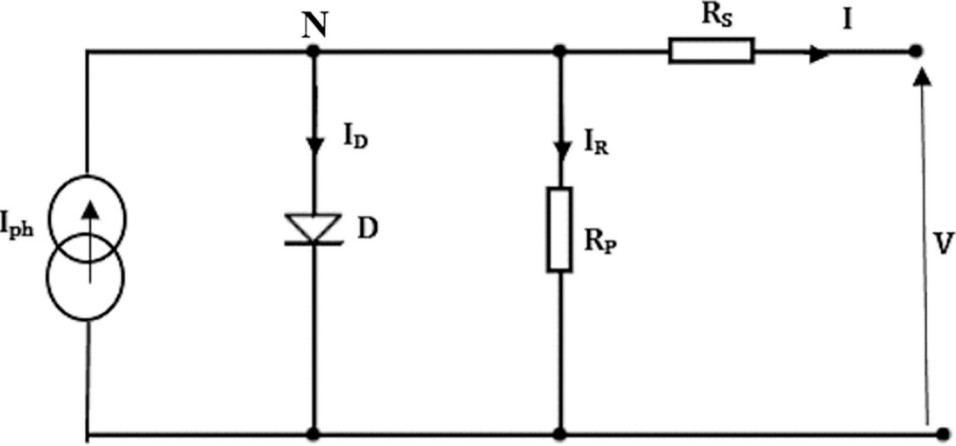

**Fig 3. Equivalent circuit of the photovoltaic cell.**

The current in the cell is maximum during the short circuit ($V = 0, I = I_{SC}$), Eq (4) becomes:

$$I_{SC} = \frac{I_{ph}}{1 + \frac{R_s}{R_P}}$$

(5)

In the ideal case ($R_S \approx 0, R_P \approx \infty$) the short-circuit current becomes:

$$I_{SC} = I_{ph}$$

(6)

## 2.2 Modeling of boost converter

The converter is modeled after analysis of the different operating sequences of the switches, the durations of which are fixed by the command [61].

**2.2.1 Single cell boost converter.** Fig 4 represents the structure of the single-cell Boost converter where T is a switch controlled by the signal $S_C$.

The differential equations system given by Eq (7), represents the state equation of the converter.

$$\begin{cases} \dfrac{dI_L}{dt} = \dfrac{V_{PV}}{L} - (1 - S_c)\dfrac{V_S}{L} \\ \dfrac{dV_S}{dt} = \dfrac{I_L}{C_2}(1 - S_c) - \dfrac{V_S}{RC_2} \end{cases}$$

(7)

The state equation of the boost converter being non-linear, its affine form can be written:

$$\dot{X} = fX + g_1 U_1 + g_2 U_2$$

(8)

with:

$X = [I_L V_S]^T$ The state vector;

$U_1, U_2$: Discontinues command;

$$\text{The state matrix } f = \begin{bmatrix} 0 & -\dfrac{1}{L} \\ \dfrac{1}{C_2} & -\dfrac{1}{RC_2} \end{bmatrix}$$

$$g_1 = \begin{bmatrix} \dfrac{V_S}{L} \\ -\dfrac{I_L}{C_2} \end{bmatrix}, \quad g_2 = \begin{bmatrix} \dfrac{V_{PV}}{L} \\ 0 \end{bmatrix}$$

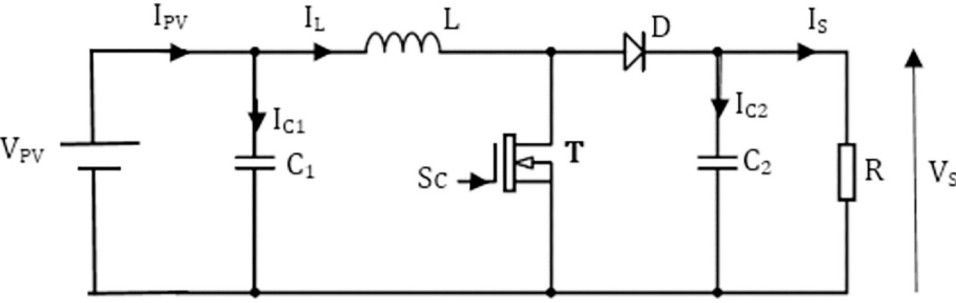

**Fig 4. One-cell boost converter.**

According to [62], the system is controllable if $det[g_1 g_2] \neq 0$

$$det[g_1 \ g_2] = det \begin{bmatrix} \dfrac{V_S}{L} & \dfrac{V_{PV}}{L} \\ -\dfrac{I_L}{C_2} & 0 \end{bmatrix} = \dfrac{I_L V_{PV}}{LC_2} \qquad (9)$$

**2.2.2 Multicellular boost converter.** In reality, static converters can only provide a chopped voltage (or current), due to the qualification of the power electronics as forced or natural switching electronics [57]. To reduce the undesirable effects of the output voltage chopping, and thus move a little more towards the "ideal converter", solutions such as increasing the number of levels available at the output of static converter, increasing the switching frequency of the output voltage so as to push the switching harmonics further and optimization of the control strategy so as to ensure the best possible tracking of the reference signal had been adopted. However, putting several switching cells in parallel presents itself as a better solution for reducing the undesirable effects of the output current chopping (Fig 2), this through to the magnetic coupler which acts as a filter by only allows pass the current that harmonics content is multiple of the cells number connected in parallel (S3 Fig) [63,64], Fig 5 shows the cases where 3 and 5 cells are connected in parallel.

The state equation of the converter made up of P cells put in parallel described in Fig 6 is given by the differential equation system described by Eq (10). For operation in multi-phase or

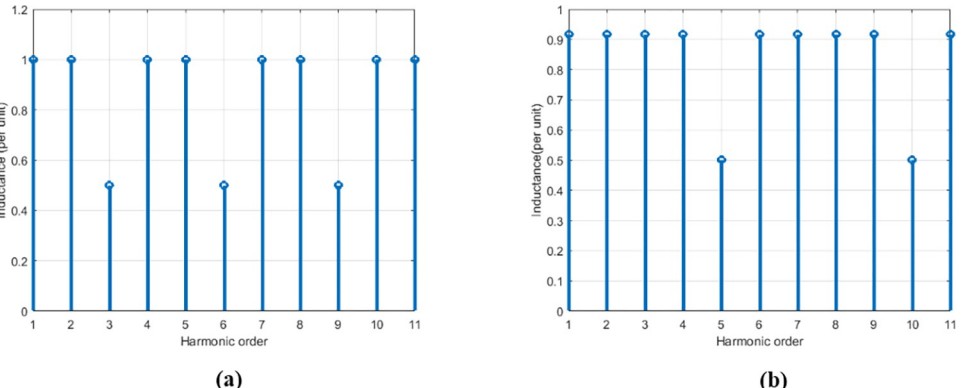

(a)                                        (b)

**Fig 5. Magnetics coupler behavior for various harmonic order.** (a) 3 cells and (b) 5 cells connected in parallel.

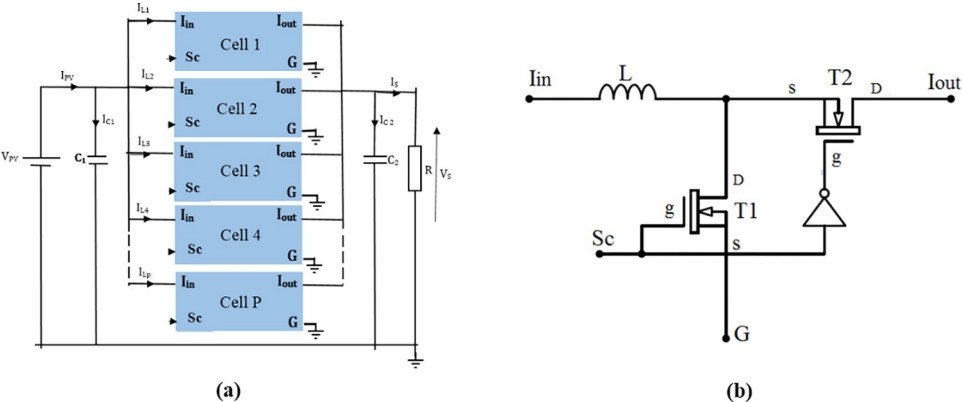

(a)                                        (b)

**Fig 6. Parallel multicellular converter with P switching cells.** (a) P switching cells. (b)Internal structure of each cell.

interleaving mode, the control signals must be shifted from each other by $Delay = \frac{period}{P}$, the current in each branch must be value $I_{branch} = \frac{1}{P}\sum_{n=1}^{P} I_{Ln}$.

$$\begin{cases} \dfrac{dI_{L1}}{dt} = \dfrac{V_{PV}}{L} - (1 - S_{c1})\dfrac{V_S}{L} \\[2mm] \dfrac{dI_{L2}}{dt} = \dfrac{V_{PV}}{L} - (1 - S_{c2})\dfrac{V_S}{L} \\[1mm] \qquad\qquad . \\ \qquad\qquad . \\ \qquad\qquad . \\ \dfrac{dI_{LP}}{dt} = \dfrac{V_{PV}}{L} - \left(1 - S_{cp}\right)\dfrac{V_S}{L} \\[2mm] \dfrac{dV_S}{dt} = \dfrac{1}{C_2}\left(\sum_{n=1}^{P} a_n I_{Ln} - \dfrac{V_S}{R}\right) \end{cases} \tag{10}$$

with $a_n = 1 - S_{cn}$,
Its affine form is:

$$\dot{X} = fX + g_1 U_1 + g_2 U_2 + \cdots + g_P U_P + g_{P+1} U_{P+1} \tag{11}$$

with:
$f$: state matrix;
$X = [I_{L1} I_{L2} \ldots I_{LP} V_S]^T$ The state vector;
$U_1, U_2, \ldots, U_P, U_{P+1}$: discontinues command.
For 5 cells connected in parallel, the state equation is given by Eq (12):

$$\begin{cases} \dfrac{dI_{L1}}{dt} = \dfrac{V_{PV}}{L} - (1 - S_{c1})\dfrac{V_S}{L} \\[2mm] \dfrac{dI_{L2}}{dt} = \dfrac{V_{PV}}{L} - (1 - S_{c2})\dfrac{V_S}{L} \\[2mm] \dfrac{dI_{L3}}{dt} = \dfrac{V_{PV}}{L} - (1 - S_{c3})\dfrac{V_S}{L} \\[2mm] \dfrac{dI_{L4}}{dt} = \dfrac{V_{PV}}{L} - (1 - S_{c4})\dfrac{V_S}{L} \\[2mm] \dfrac{dI_{L5}}{dt} = \dfrac{V_{PV}}{L} - (1 - S_{c5})\dfrac{V_S}{L} \\[2mm] \dfrac{dV_S}{dt} = \dfrac{I_{L1}}{C_2}(1 - S_{c1}) + \dfrac{I_{L2}}{C_2}(1 - S_{c2}) + \dfrac{I_{L3}}{C_2}(1 - S_{c3}) + \dfrac{I_{L4}}{C_2}(1 - S_{c4}) + \dfrac{I_{L5}}{C_2}(1 - S_{c5}) - \dfrac{V_S}{RC_2} \end{cases} \tag{12}$$

$$\dot{X} = fX + g_1 U_1 + g_2 U_2 + g_3 U_3 + g_4 U_4 + g_5 U_5 + g_6 U_6 \tag{13}$$

$X = [I_{L1} I_{L2} I_{L3} I_{L4} I_{L5} V_S]^T$ the state vector;

$U_1, U_2, U_3, U_4, U_5, U_6$: Discontinues command;

$$f = \begin{bmatrix} 0 & 0 & 0 & 0 & 0 & -\dfrac{1}{L} \\ 0 & 0 & 0 & 0 & 0 & -\dfrac{1}{L} \\ 0 & 0 & 0 & 0 & 0 & -\dfrac{1}{L} \\ 0 & 0 & 0 & 0 & 0 & -\dfrac{1}{L} \\ 0 & 0 & 0 & 0 & 0 & -\dfrac{1}{L} \\ \dfrac{1}{C_2} & \dfrac{1}{C_2} & \dfrac{1}{C_2} & \dfrac{1}{C_2} & \dfrac{1}{C_2} & -\dfrac{1}{RC_2} \end{bmatrix}$$

$$g_1 = \begin{bmatrix} \dfrac{V_S}{L} \\ 0 \\ 0 \\ 0 \\ 0 \\ -\dfrac{I_{L1}}{C_2} \end{bmatrix}, g_2 = \begin{bmatrix} 0 \\ \dfrac{V_S}{L} \\ 0 \\ 0 \\ 0 \\ -\dfrac{I_{L2}}{C_2} \end{bmatrix}, g_3 = \begin{bmatrix} 0 \\ 0 \\ \dfrac{V_S}{L} \\ 0 \\ 0 \\ -\dfrac{I_{L3}}{C_2} \end{bmatrix}, g_4 = \begin{bmatrix} 0 \\ 0 \\ 0 \\ \dfrac{V_S}{L} \\ 0 \\ -\dfrac{I_{L4}}{C_2} \end{bmatrix}, g_5 = \begin{bmatrix} 0 \\ 0 \\ 0 \\ 0 \\ \dfrac{V_S}{L} \\ -\dfrac{I_{L5}}{C_2} \end{bmatrix}, g_6 = \begin{bmatrix} \dfrac{V_{PV}}{L} \\ \dfrac{V_{PV}}{L} \\ \dfrac{V_{PV}}{L} \\ \dfrac{V_{PV}}{L} \\ \dfrac{V_{PV}}{L} \\ \dfrac{V_{PV}}{L} \\ 0 \end{bmatrix}$$

Controllability:

$$det[g_1\ g_2\ g_3\ g_4\ g_5\ g_6] = \frac{V_{PV}V_S}{L^5 C_2}(I_{L1} + I_{L2} + I_{L3} + I_{L4} + I_{L5}) \tag{14}$$

$det[g_1 g_2 g_3 g_4 g_5 g_6] \neq 0$, then the system is controllable.

## 2.3 Method based on control of the inductance current

**2.3.1 Principle of the method.**   The diagram of the method is illustrated in Fig 7.

According to Eqs (9) and (14), the system can be control from the current flowing through the inductance. Fig 7 gives according to KCL, the relationship linking the PV generator current ($I_{pv}$), inductance current ($I_L$) and the capacitor current ($I_C$):

$$I_{pv} = I_L + I_{C1} \tag{15}$$

$$Où\ \ I_{C1} = C_1 \frac{dV_{C1}}{dt} = C_1 \frac{dV_{pv}}{dt} \approx C_1 \frac{\Delta V_{pv}}{\Delta T} \tag{16}$$

In steady state or for small variations $V_{pv}$ Voltage, Eq (15) becomes:

$$I_{pv} \approx I_L \tag{17}$$

From Eqs (17) and (6), we will choose the reference current by estimating the photocurrent due to the high influence of irradiance on the output current from GPV. The relationship

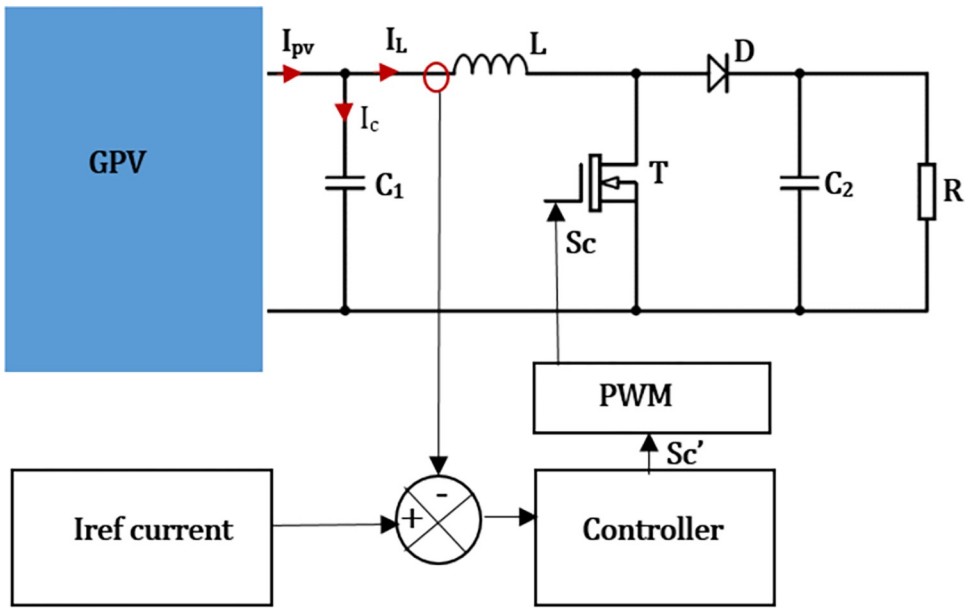

**Fig 7. Basic diagram of the method by controlling the inductance current.**

linking the photocurrent to the irradiance and temperature is given by [61,65]:

$$I_{ph} = [I_{SC} + \alpha(T_C - T_r)]\left(\frac{G}{G_r}\right) \qquad (18)$$

Where $\alpha$ is the short circuit current temperature coefficient (A/K), $T_C$ is the temperature of the cell (K), $T_r$ is the reference temperature (25°C or 298K), $G$ is the irradiance ($W/m^2$) and $G_r$ the reference irradiance ($1000/m^2$).

$$\alpha = \frac{dI_{SC}}{dT_C} \qquad (19)$$

When $T_C = T_r$,

$$I_{ph} = I_{SC}\left(\frac{G}{G_r}\right)$$

According to [66] for a polycrystalline silicon material, $\alpha = 0{,}0021 A/K$; according to [67] for a silicon material, $\alpha = 0{,}00238 A/K$ for polycrystalline structure and $0{,}00175 A/K$ for the single crystal structure. Given these values, Eq (18) can be expressed:

$$I_{ph} = I_{SC}\left(\frac{G}{G_r}\right) \qquad (20)$$

In control technics based on the proportionality relation of the linear relation in first approach between $I_{OPT}$ and $I_{SC}$ described by [68,69],

$$I_{OPT} = K_I I_{SC} \qquad (21)$$

$$I_{SC} = \frac{I_{OPT}}{K_I} \qquad (22)$$

Eq (22) in Eq (20) give the following relation:

$$I_{ph} = \frac{I_{OPT}}{K_I}\left(\frac{G}{G_r}\right) \tag{23}$$

$K_I$ being a current factor generally between 0,78 $et$ 0,92; let's put $\frac{1}{K_I} \approx 1$, then Eq (23) can be written:

$$I_{ph,ref} \approx I_{OPT}\left(\frac{G}{G_r}\right)$$

The reference current can be formula in the form

$$I_{ref} = I_{ph,ref} = K_G I_{OPT} \tag{24}$$

**2.3.2 Controller.** To ensure control of current through the inductance, the controller whose goal is to oblige the inductance current to follow the reference current by automatically updating the corresponding duty cycle is built around PI regulator whose the transfer function is given by Eq (25):

$$C(s) = K_P + \frac{K_I}{s} \tag{25}$$

From Fig 7,

$$\varepsilon(s) = I_{ref}(s) - I_L(s) \tag{26}$$

$$S'(s) = \varepsilon(s)C(s) \tag{27}$$

The parameters $K_P$ $and$ $K_I$ must be chosen so as to make if possible $\varepsilon(s)\approx0°$; for this the particle swarm optimization (PSO) algorithm which the objective function shown on Fig 8 and

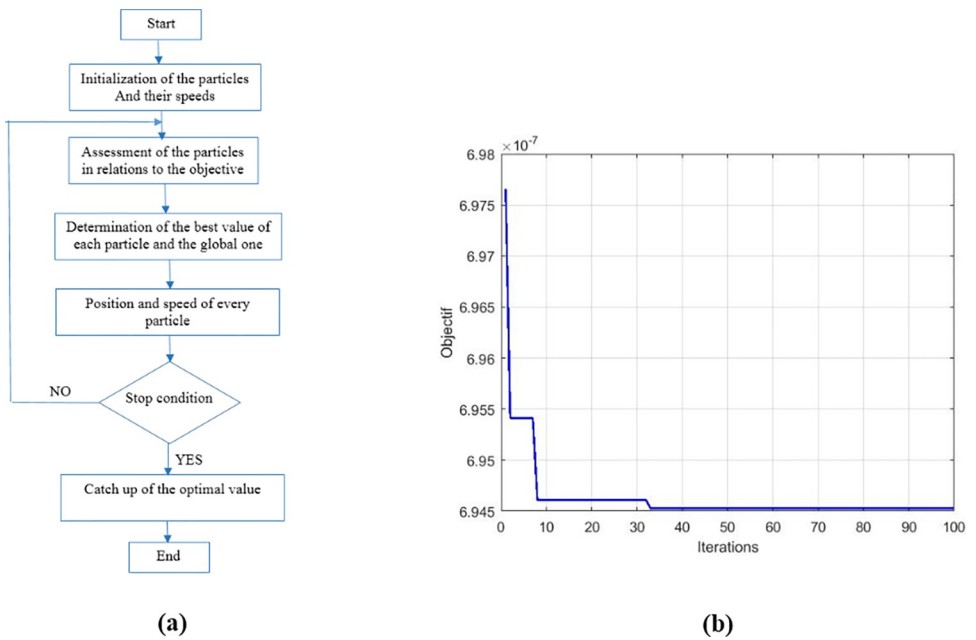

(a)                                              (b)

**Fig 8. PSO characteristics.** (a)PSO flowchart. (b)Objective function.

describe by Eq (28) was used for choose the values of $K_P$ and $K_I$.

$$\begin{cases} V_{i+1} = \mu_1 V_i + \mu_2(x_{ip} - x_i) + \mu_3(x_g - x_i) \\ x_{i+1} = x_i + V_{i+1} \end{cases} \tag{28}$$

Where,

$V_{i+1}, x_{i+1}$ are respectively the updated speed and position of each particle $i$;

$V_i, x_i$ are respectively the actual speed and position of particle $i$;

$\mu_1, \mu_2, \mu_3 \in [0,1]$; $x_{ip}, x_g$ are respectively the best position visited by the particle $i$ and the best position visited by the swarm.

With parameter values: Number of iterations = 100; Number of particle s = 100; Speed_-max = 2.5; Speed_min = -2.5; p_best_init = 100000000; g_best_init = 100000000; Kp_min = 0; Kp_max = 2000;

Ki_min = 0; Ki_max = 2000;

## 2.4 Method of control by sliding mode

The diagram of control by sliding mode illustrated in Fig 9, shows that the reference current generated previously can be used as reference current in sliding mode control. The Lyapunov criterion is used to define switching functions because of the simplicity of its implementation. The quadratic Lyapunov function is define around energies stored in capacitors and inductances as follow:

$$V = \frac{1}{2} \Delta X^T . H . \Delta X \tag{29}$$

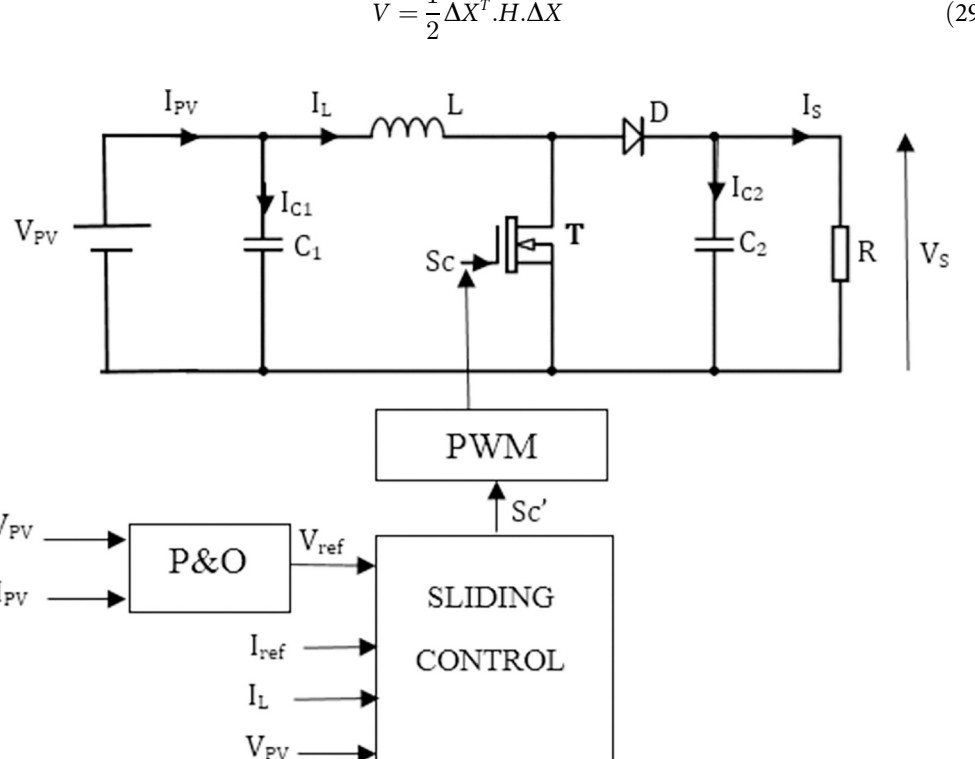

**Fig 9. Diagram of control by sliding mode.**

Where H is a diagonal constant matrix containing inductive and capacitive elements;

$$H = \begin{bmatrix} L_1 & 0 & \dots & 0 & 0 \\ 0 & L_2 & \dots & 0 & 0 \\ \vdots & \vdots & \ddots & \vdots & \vdots \\ 0 & 0 & 0 & L_P & 0 \\ 0 & 0 & 0 & 0 & C_2 \end{bmatrix}$$

According to Eqs (9) and (14), we pose the error $\Delta X = X_{ref} - X$ as follow:

$$\Delta X = \begin{bmatrix} I_{ref} - I_{L1} \\ I_{ref} - I_{L2} \\ \vdots \\ I_{ref} - I_{Lp} \\ V_{ref} - V_{pv} \end{bmatrix}$$

The system will be stable in a closed loop if the derivative of the Lyapunov function is negative,

$$\dot{V} = \Delta X^T . H . \Delta \dot{X} \tag{30}$$

$$\Delta \dot{X} = \dot{X} = fX + g_1 U_1 + g_2 U_2 + \dots + g_P U_P + g_{P+1} U_{P+1} \tag{31}$$

Switching functions are defined through the following relationship:

$$S_i = -\Delta X^T . H . g_i, \qquad i = 1, 2, \dots p, p + 1 \tag{32}$$

For P = 5,

$$H = \begin{bmatrix} L_1 & 0 & 0 & 0 & 0 & 0 \\ 0 & L_2 & 0 & 0 & 0 & 0 \\ 0 & 0 & L_3 & 0 & 0 & 0 \\ 0 & 0 & 0 & L_4 & 0 & 0 \\ 0 & 0 & 0 & 0 & L_5 & 0 \\ 0 & 0 & 0 & 0 & 0 & C_2 \end{bmatrix}$$

$$\Delta X = \begin{bmatrix} I_{ref} - I_{L1} \\ I_{ref} - I_{L2} \\ I_{ref} - I_{L3} \\ I_{ref} - I_{L4} \\ I_{ref} - I_{L5} \\ V_{ref} - V_{pv} \end{bmatrix}$$

$$\Delta \dot{X} = \dot{X} = fX + g_1 U_1 + g_2 U_2 + g_3 U_3 + g_4 U_4 + g_5 U_5 + g_6 U_6$$

$$S_i = -\Delta X^T . H . g_i, \qquad i = 1, 2, 3, 4, 5, 6$$

$$S_1 = (V_{ref} - V_{pv})I_{L1} - (I_{ref} - I_{L1})V_S$$

$$S_2 = (V_{ref} - V_{pv})I_{L2} - (I_{ref} - I_{L2})V_S$$

$$S_3 = (V_{ref} - V_{pv})I_{L3} - (I_{ref} - I_{L3})V_S$$

$$S_4 = (V_{ref} - V_{pv})I_{L4} - (I_{ref} - I_{L4})V_S$$

$$S_5 = (V_{ref} - V_{pv})I_{L5} - (I_{ref} - I_{L5})V_S$$

$$S_6 = -V_{pv}[5I_{ref} - (I_{L1} + I_{L2} + I_{L3} + I_{L4} + I_{L5})]$$

Error $\Delta X$ is stable if $S_i = 0°$; for $S_6 = 0$, we obtain:

$$I_{ref} = \frac{I_{L1} + I_{L2} + I_{L3} + I_{L4} + I_{L5}}{5}$$

Thus giving the current to pass through each branch for multicellular use.

## 2.5 Calculation of efficiency

According to [17], the calculation of the efficiency in a photovoltaic chain is product of the MPPT efficiency ($\eta_{MPPT}$) and the conversion efficiency ($\eta_{CONV}$) of the static converter. The MPPT efficiency determines the effectiveness of control technics in terms for tracking the maximum power point; it is given by the following relation:

$$\eta_{MPPT} = \frac{P_{pv}}{P_{MPP}} \tag{33}$$

Where $P_{pv}$ is the power delivered by the photovoltaic generator, $P_{MPP}$ the maximum power of the photovoltaic generator at the maximum power point.

The efficiency of a static converter can be defined as its ability to transfer at its output the maximum of available power at its input [49]:

$$\eta_{CONV} = \frac{P_{out}}{P_{pv}} = \frac{P_{out}}{P_{out} + P_{loss}} = \frac{1}{1 + \frac{P_{loss}}{P_{out}}} \tag{34}$$

$P_{pv}$ is the power delivered by the photovoltaic generator which becomes the input power of converter, $P_{out}$ is the power transferred to the converter output.

The total efficiency of the chain is given by:

$$\eta_{CHAIN} = \eta_{MPPT} \times \eta_{CONV} \tag{35}$$

## 3. Simulation results and discussion

In this part, we present the different simulations carried out in Matlab/Simulink environment and their results. We compared the suggested solutions based on the control of the inductance

**Table 1. Designed parameters for simulation.**

| Components | Values |
|---|---|
| $C_1 = C_2$ | $200\mu F$ |
| $L$ | $100\mu H$ |
| $R$ | $58\Omega$ |
| Switching frequency ($F_{sw}$) | $5KHz$ |
| $K_P$ | 12.73 |
| $K_i$ | 10000 |

current and multicellular firstly to some traditional MPPT technics control, and to some solutions proposed by others works in term of improvement of power quality. Simulations were carried out in various cases: under constant irradiance, variable irradiance and variable temperature. The photovoltaic module used is type Kyocera Solar KC200GT which some characteristics depending on the irradiance under a constant temperature of 25˚C are given in S1 Table and S1 Fig; Table 1 give the designed parameters for simulation.

### 3.1 functioning of conventional boost converter with the suggested MPPT for various duty cycle

The proposed MPPT was simulated on the conventional boost to check if it responds to operating criteria of this converter.

Some duty cycles were used: 0.25 (Fig 10), 0.5 (Fig 11) and 0.75 (Fig 12). for 0.5 duty cycle value, the output voltage of the boost converter must be equal to twice the value of its input voltage, this is verified by Fig 11.

### 3.2 Comparison with some MPPT technics under constant irradiance $G = 1000\,W/m^2$ and variable irradiance for temperature $T = 25˚C$

The proposed solutions are compared to some MPPT technics such as: perturb and observe (P&O) in Fig 13, fraction short circuit (FSCC) in Fig 14, fuzzy logic (Fig 15) and hybrid sliding mode assisted by P&O in Fig 16. The operations were simulated under the conditions of constant irradiance, variable irradiance according to the profile of Fig 17, both under a constant temperature of 25˚C.

To make the comparison between the different maximum point search techniques in terms of speed, stability, precision, an analysis was carried out and grouped in Table 2, and in

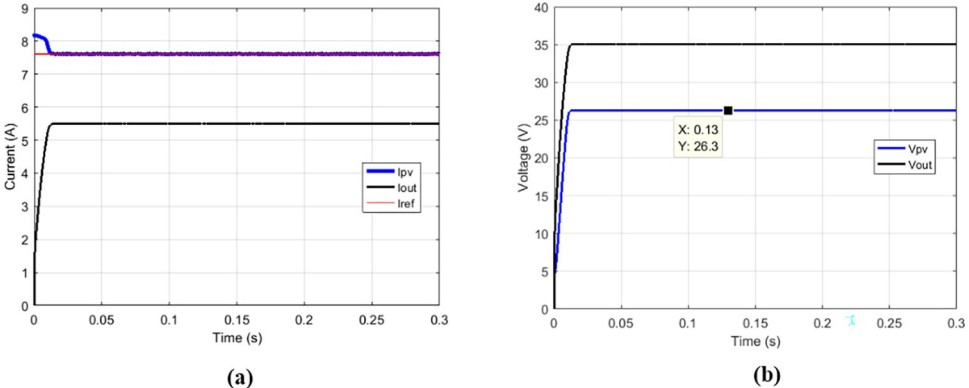

(a) (b)

**Fig 10. Conventional boost with suggested MPPT for duty cycle = 0.25.** (a)Currents. (b)Voltages.

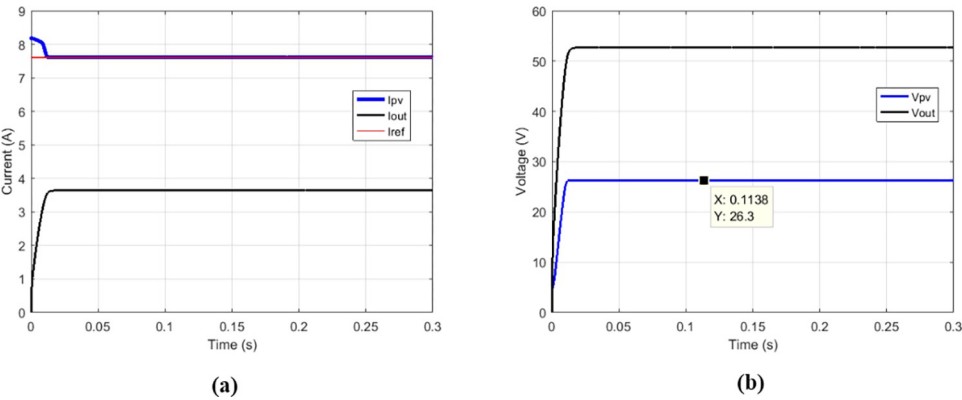

**Fig 11. Conventional boost with suggested MPPT for duty cycle = 0.5.** (a)Currents. (b)Voltages.

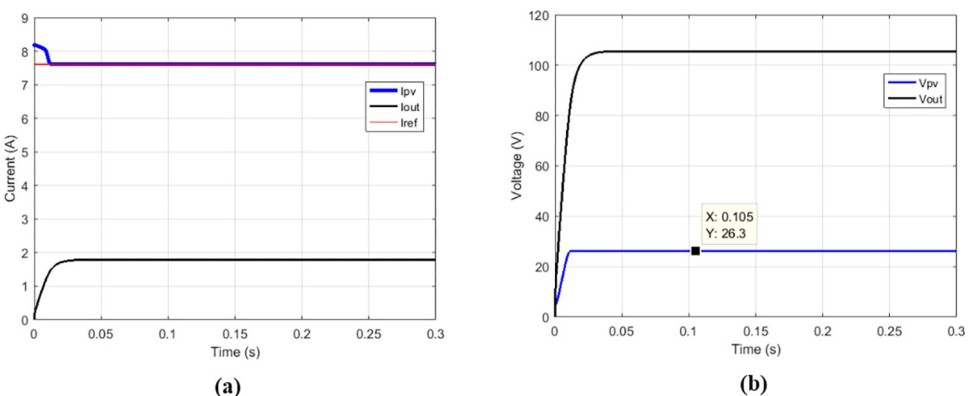

**Fig 12. Conventional boost with suggested MPPT for duty cycle = 0.75.** (a)Currents. (b)Voltages.

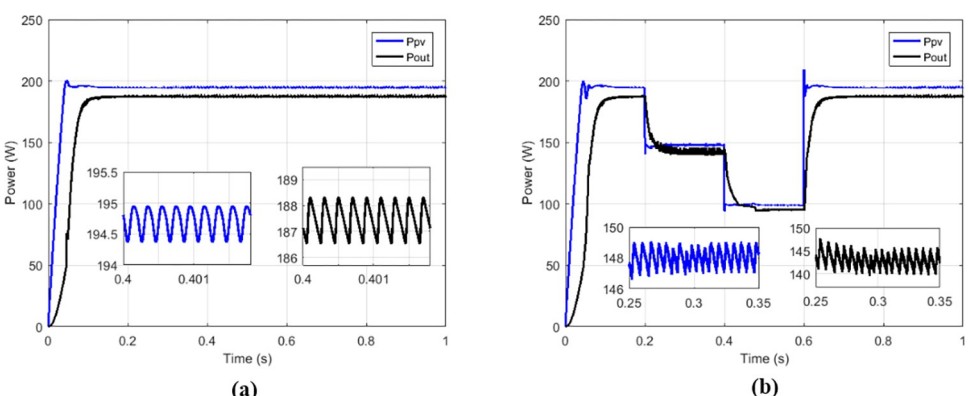

**Fig 13. Powers curves with conventional boost based on P&O MPPT.** (a)Constant irradiance. (b)Variable irradiance.

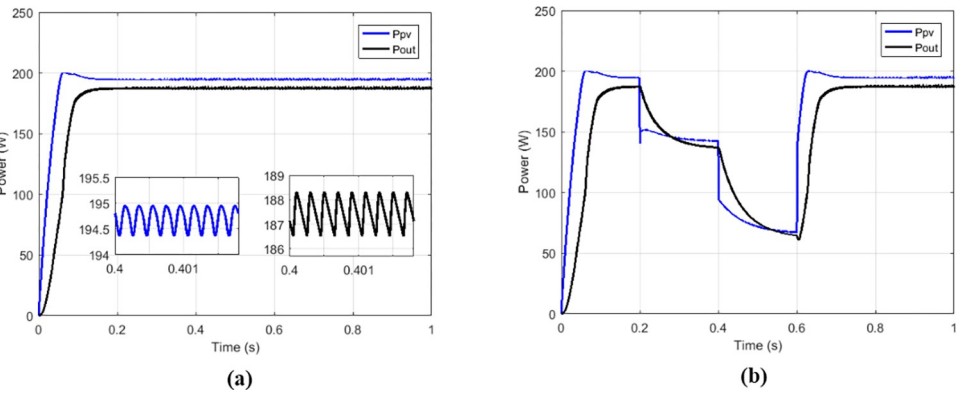

**Fig 14. Powers curves with conventional boost based on FSCC MPPT.** (a)Constant irradiance. (b)Variable irradiance.

Table 3 for the comparison in terms of transferred power and efficiency. From the simulation results we notice that the fuzzy MPPT technique (Fig 15A) presents very good stability under constant irradiance compared to the others, but when faced with variations it tends to lose tracking. the sliding control (Fig 16) used here according to Fig 9 is a hybrid control combining the sliding mode assisted by the P&O control. In Fig 18 the proposed MPPT was used to control the conventional boost. The proposed solution which consists to use the proposed MPPT and multicellular converter (Fig 19), despite the use of low values of the filtering elements (C1, C2 and L), presents better results in terms of tracking, stability, precision and power transfer.

## 3.3 Simulation of the proposed solution under temperature variations at constant irradiance $G = 1000W/m^2$

The proposed solution was subjected to a temperature variation that the profile is illustrated on Fig 20, we notice in Fig 21A a very weak influence of the temperature on the current, according to Fig 22B we observe the tracking of the maximum point of the power of the GPV with a response time of 0.002 seconds and a response time of 0.02 seconds for the transferred power. These results agree with the values described in S2 Table and S2 Fig.

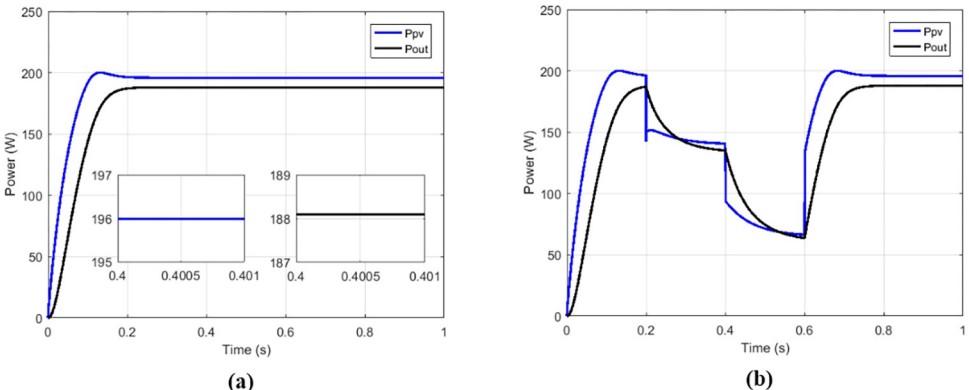

**Fig 15. Powers curves with conventional boost based on fuzzy MPPT.** (a)Constant irradiance. (b)Variable irradiance.

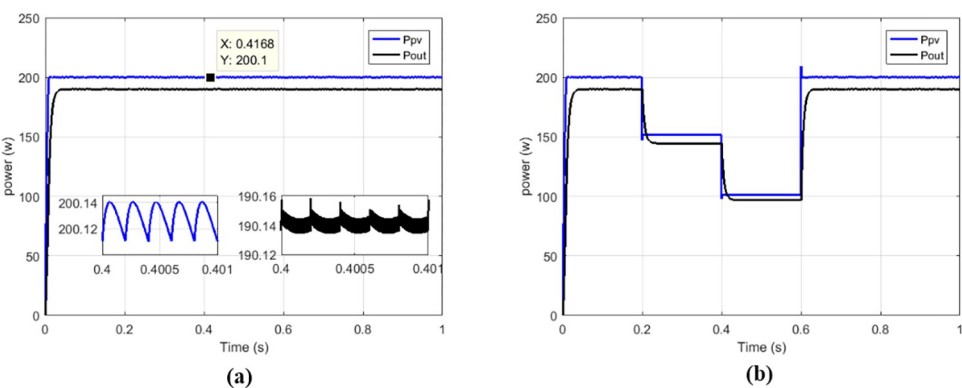

**Fig 16. Powers curves with conventional boost based on sliding mode MPPT.** (a)Constant irradiance. (b)Variable irradiance.

## 3.4 Comparison of proposed solution with HG-QBC

The proposed solution is compared to the high-gain quadratic boost converter (HG-QBC) proposed by [43] in order to appreciate its performances. The Figs 23–26 show that the proposed solution can operate in a wide voltage range.

Fig 26B show more the behaviors facing the different change of variations according to profile presented on Fig 23A. An analysis is made between the performances of the proposed solution and the performances of the HG-QBC proposed by [43] and is presented in Tables 4 and 5.

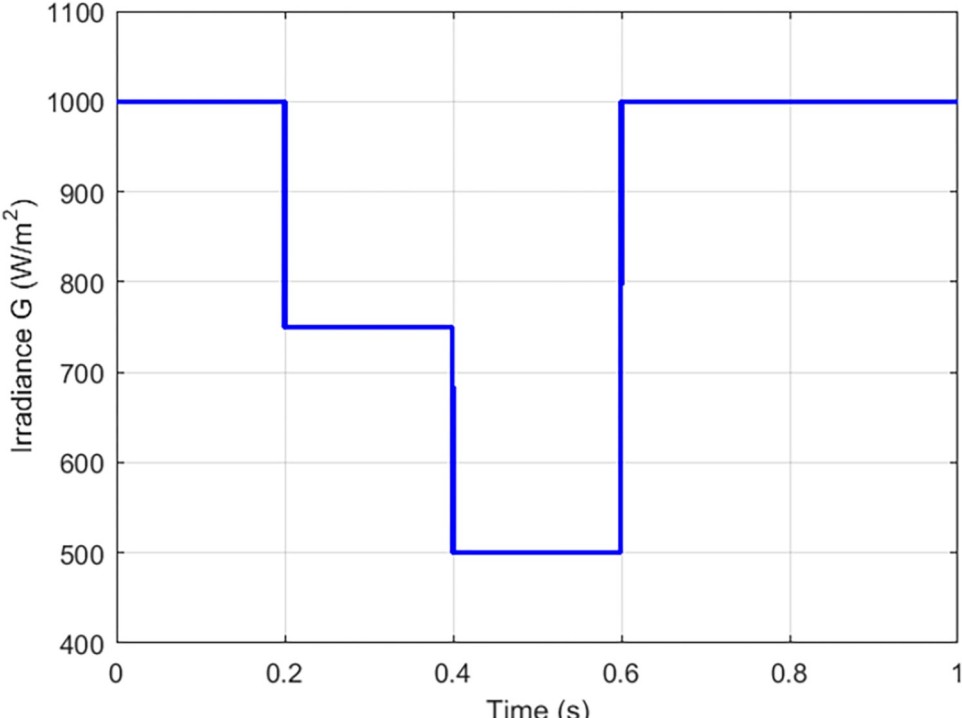

**Fig 17. Irradiance profile under variation conditions.**

**Table 2. Comparison among some solutions in terms of powers, response times and oscillations.**

| Type of MPPT | Type of converter | $P_{MPP}$ | $P_{pv}$ | $T_{r\_pv}$ | $P_{pv}$ oscillations |
|---|---|---|---|---|---|
| P&O | Conventional boost | 200.14 W | 194.5 W | 163.1 ms | 0.58 W |
| FSC | Conventional boost | 200.14 W | 194.4 W | 152.5 ms | 0.58 W |
| Fuzzy | Conventional boost | 200.14 W | 196 W | 250 ms | 19.1 μW |
| Sliding | Conventional boost | 200.14 W | 200.1 W | 10.2 ms | 0.07 W |
| Proposed | Conventional boost | 200.14 W | 200.1 W | 11.1 ms | 0.24 W |
| Proposed | Multicellular | 200.14 W | 2001. W | 10 ms | 0.05 W |

Where $P_{MPP}$ is maximum power of GPV at MPP, $P_{pv}$ is power delivered at out of GPV, $T_{r\_PV}$ is response time.

**Table 3. Comparison among some solutions in terms of transferred powers and efficiencies.**

| Type of MPPT | Type of converter | $P_{pv}$ | $P_{out}$ | $\eta_{MPPT}$ | $\eta_{CONV}$ | $\eta_{CHAIN}$ |
|---|---|---|---|---|---|---|
| P&O | Conventional | 194.5 W | 187 W | 97.18% | 96.14% | 93.43% |
| FSC | Conventional | 194.4 W | 187.5 W | 97.13% | 96.45% | 93.68% |
| Fuzzy | Conventional | 196 W | 188 W | 97.93% | 95.91% | 93.93% |
| Sliding | Conventional | 200.1W | 190.4 W | 99.98% | 95.15% | 95.13% |
| Proposed | Conventional | 200.1 W | 188.78 W | 99.98% | 94.34% | 94.32% |
| Proposed | Multicellular | 200.1 W | 198.5 W | 99.98% | 99.20% | 99.18% |

Where $P_{out}$ is output power of the converter, $\eta_{MPPT}$ is MPPT efficiency, $\eta_{CONV}$ is conversion efficiency, $\eta_{CHAIN}$ is the total efficiency of the chain.

By analyzing values presented on Table 5 to values on S1 Table and S1 Fig for different irradiance values, we observe that the proposed solution presents best results in term of precision of tracking.

### 3.5 Comparison of proposed solution with IHGBC

Among the solutions for improving the performance of boost converters dedicated to PV applications, there is the interleaved high-gain boost converter (IHGBC) proposed by [44]. This converter has the capacity to increase the voltage gain with less ripple output. Fig 27A presents irradiance profile under variations; the currents, voltages and output power operate at constant irradiance are respectively show on Figs 27B and 28.

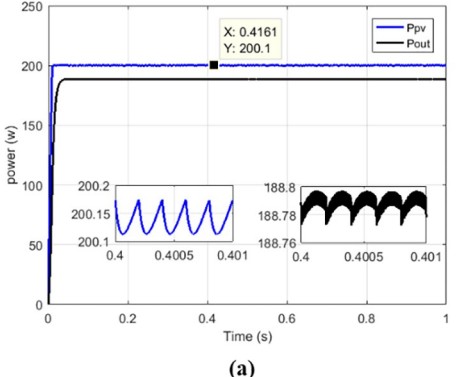
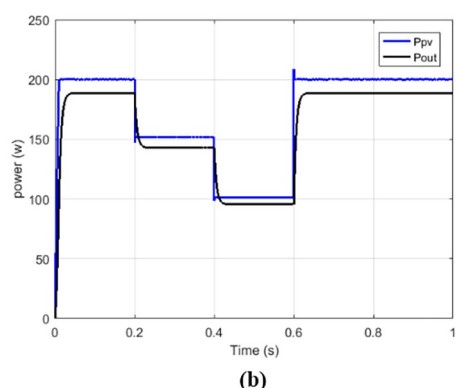

(a)                                           (b)

**Fig 18. Powers curves with conventional boost based on proposed MPPT.** (a)Constant irradiance. (b)Variable irradiance.

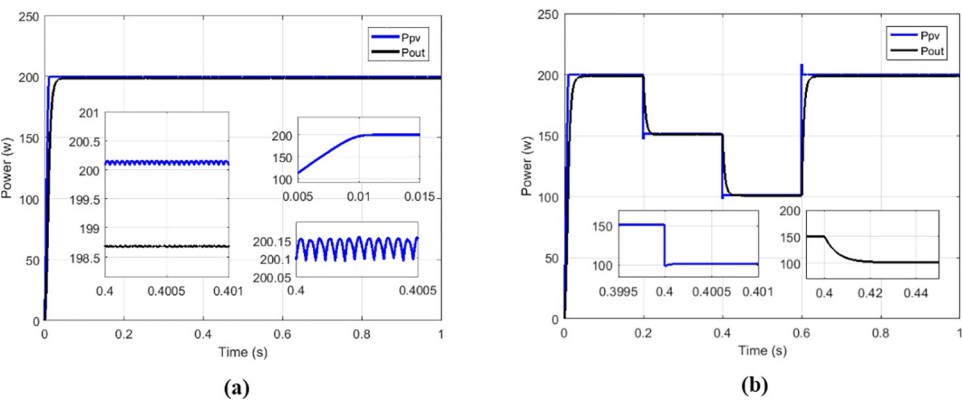

**Fig 19. Powers curves with parallel boost converter based on proposed MPPT.** (a)Constant irradiance. (b)Variable irradiance.

Under variable irradiance, currents are presented on Fig 29A, voltage on Fig 29B and powers on Fig 30. Fig 30B presents more in details the oscillations and response time of the power delivered by GPV ($P_{PV}$) and the transferred output power ($P_{out}$). Table 6 presents the comparison made between the proposed solution and the IHGBC. In spite of slow response time of proposed solution facing the IHGBC, Table 6 show best performances of the proposed solution in term of ripples minimization and maximum transferred power to output.

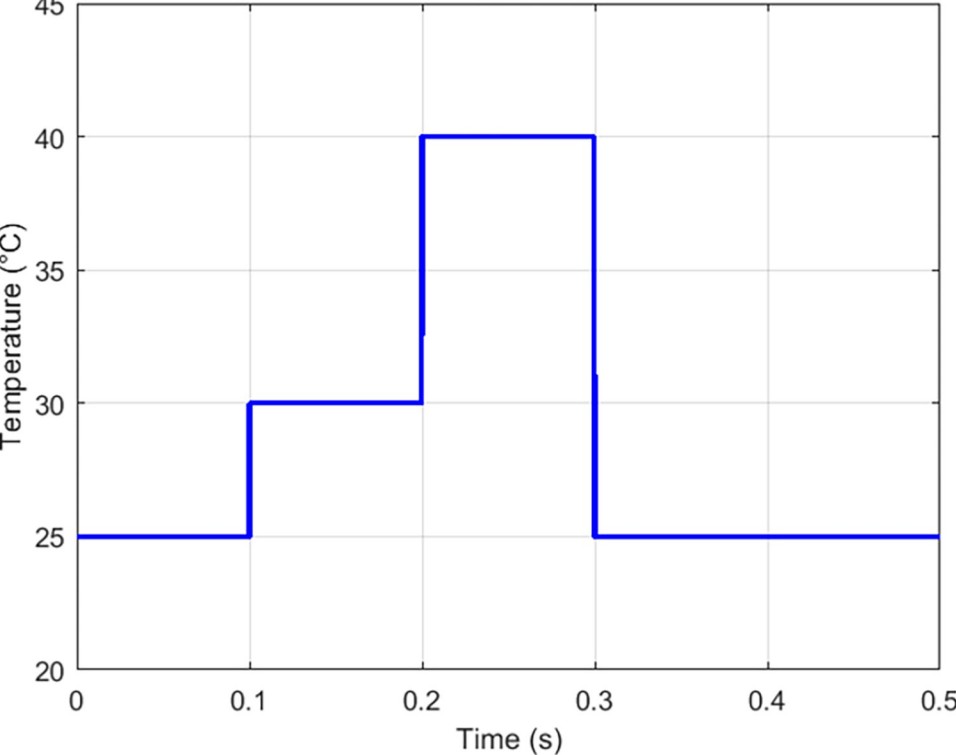

**Fig 20. Temperature profile under variation conditions.**

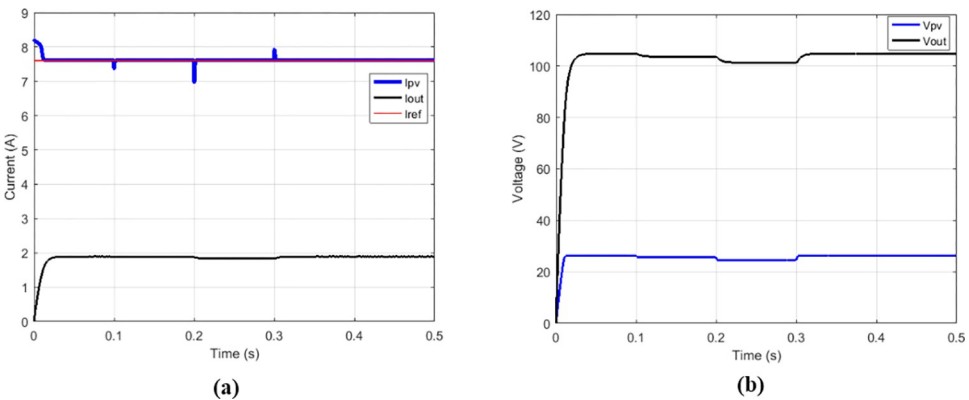

**Fig 21. Proposed solution under temperature variations at constant irradiance.** (a)Currents. (b)Voltage.

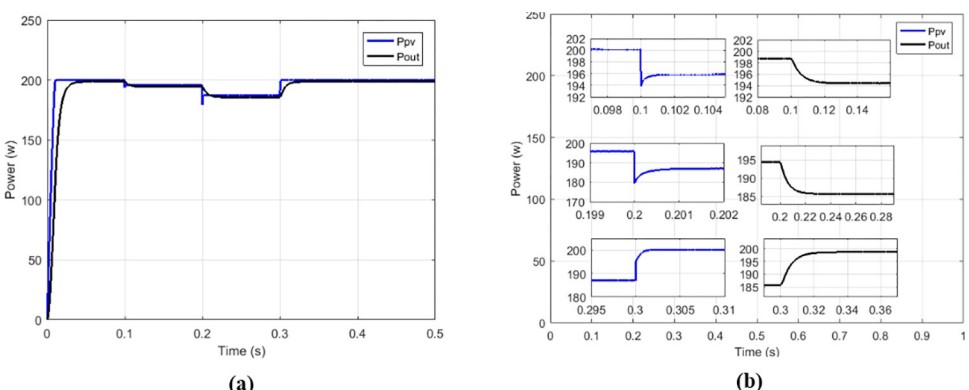

**Fig 22. Proposed solution under temperature variations at constant irradiance.** (a)Powers. (b)powers zoom.

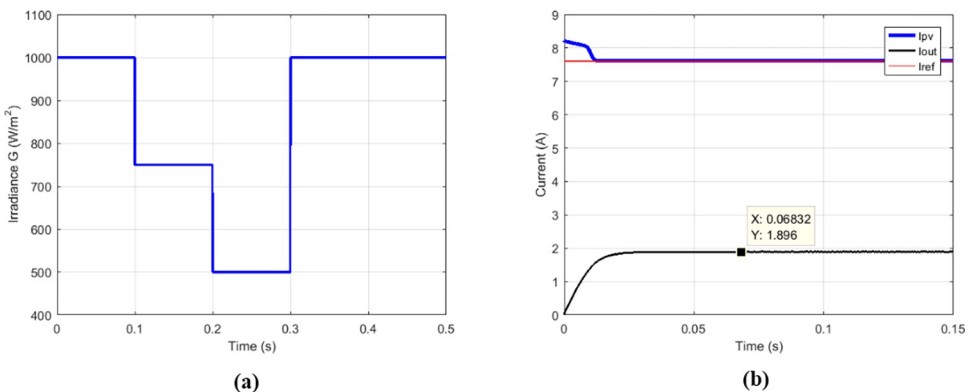

**Fig 23.** (a) Irradiance profile. (b) Currents at constant irradiance.

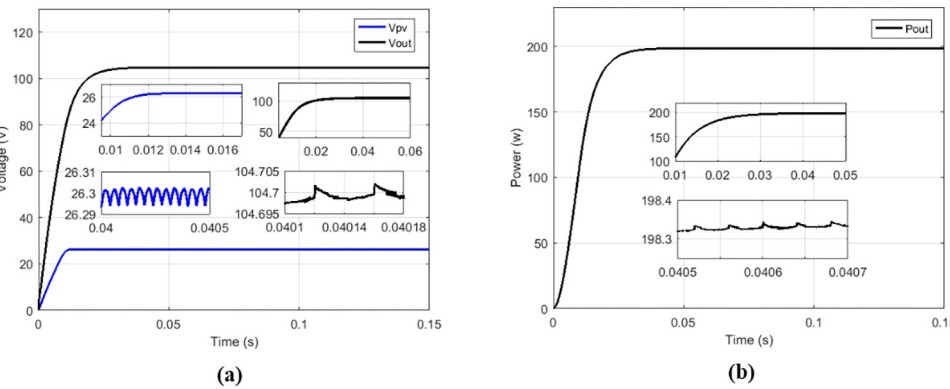

**Fig 24.** (a)Voltages and (b)output power at constant irradiance.

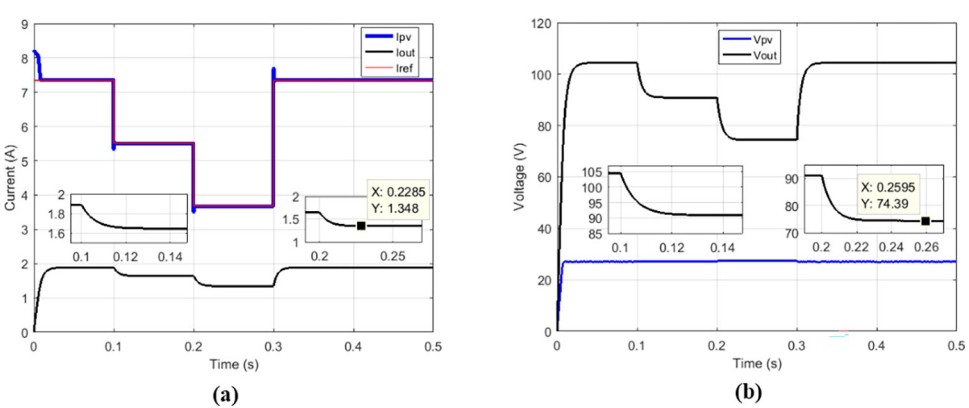

**Fig 25.** (a)Currents and (b)Voltages under variable irradiance.

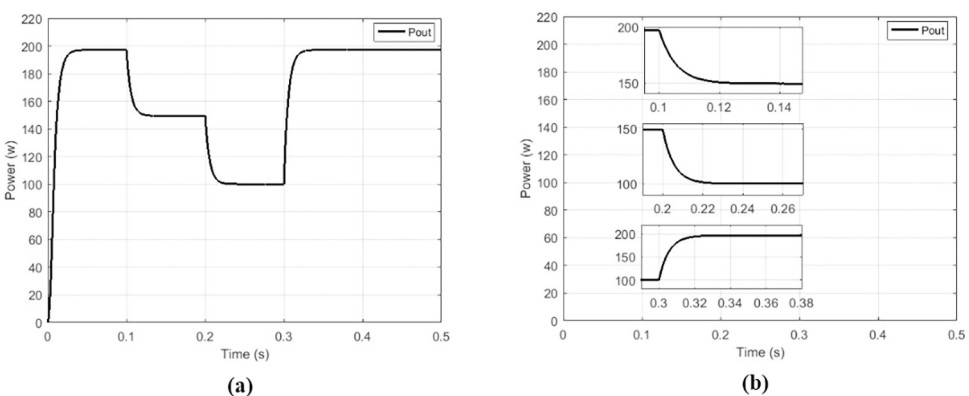

**Fig 26.** (a)Output Power and (b)output power zoom under variable irradiance.

**Table 4. Performance analysis of the proposed solution and the HG-QBC.**

| Type of MPPT | Type of converter | Setting time | $V_{out}$ | $V_{out}$ripples | Efficacy |
|---|---|---|---|---|---|
| Hybrid (PO&NN) [43] | HG-QBC | 1.2 second | 104.8 V | 0.2% | 97.5% |
| Proposed | Multicellular | 0.04 second | 104.7 V | 0.005% | 99.08% |

**Table 5. Performance analysis of the proposed solution and the HG-QBC under irradiance variation.**

| Irradiance | Hybrid (PO&NN) using HG-QBC [43] | | | Proposed MPPT using Multicellular | | |
|---|---|---|---|---|---|---|
| | $V_{out}$ | $I_{out}$ | $P_{out}$ | $V_{out}$ | $I_{out}$ | $P_{out}$ |
| $G = 500 W/m^2$ | 75.78 V | 1.316 A | 99.79 W | 74.39 V | 1.348 A | 100.1 W |
| $G = 750 W/m^2$ | 90.79 V | 1.578 A | 143.1 W | 90.91 V | 1.646 A | 149.7 W |
| $G = 1000 W/m^2$ | 104.8 V | 1.819 A | 190.63 W | 104.7 V | 1.896 A | 198.3 W |

Where $V_{out}, I_{out}, P_{out}$ are respectively output voltage, output current and output power of the converter.

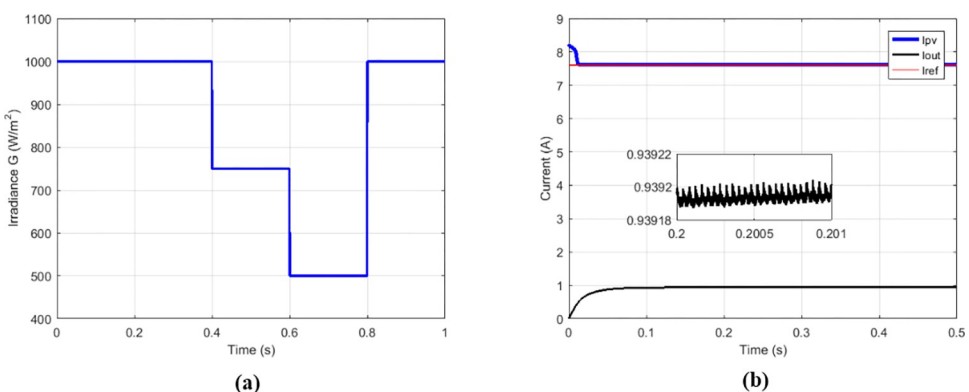

(a)                                                                (b)

**Fig 27.** (a) Irradiance profile. (b) Currents at constant irradiance.

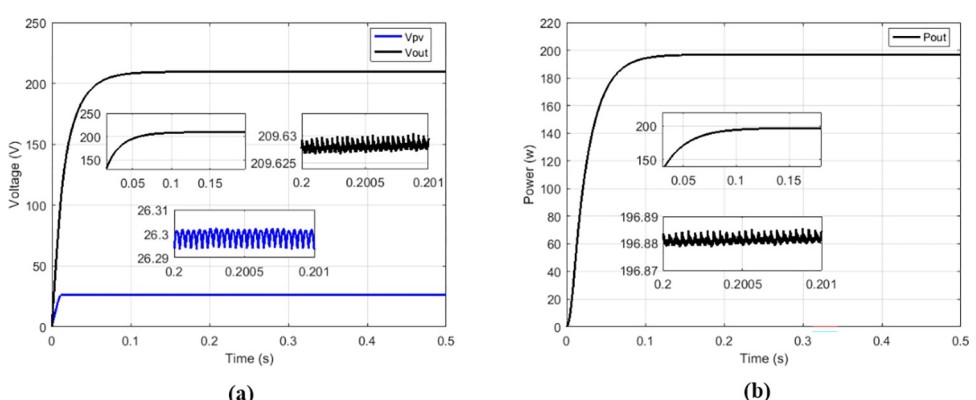

(a)                                                                (b)

**Fig 28.** (a) Voltages and (b) output power at constant irradiance.

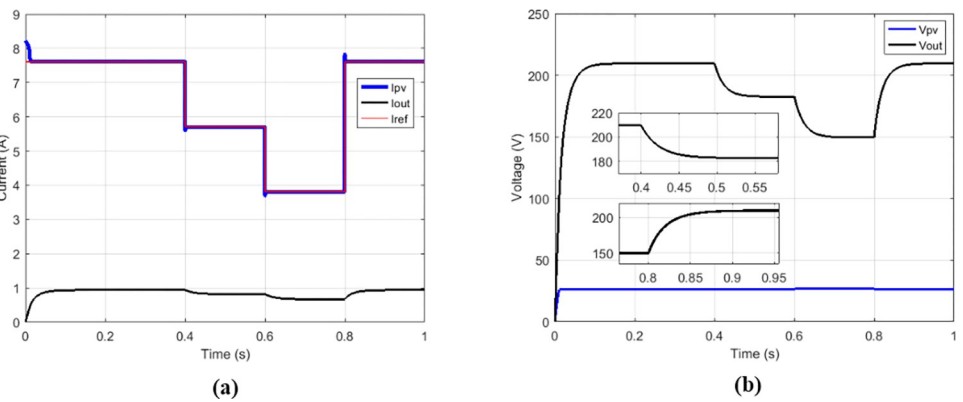

**Fig 29.** (a) Currents and (b) voltages under variable irradiance.

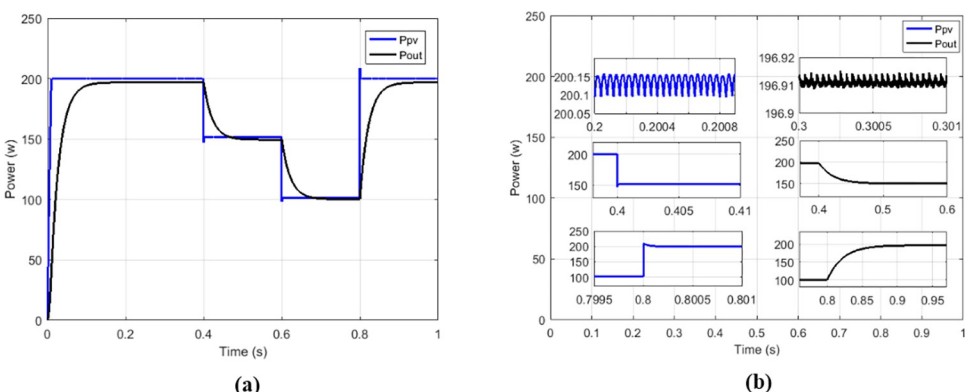

**Fig 30.** (a) Powers and (b) powers zoom under variable irradiance.

**Table 6. Performance analysis of the proposed solution and the IHGBC.**

| Magnitude | IHGBC using Hybrid (P&O-FP) MPPT [44] | Multicellular using Proposed MPPT |
|---|---|---|
| Switching frequency | 50 KHz | 5 KHz |
| $V_{out}$ | 209.4 V | 209.62 V |
| $V_{out}$ ripples | 0.14% | 0.001% |
| $I_{out}$ | 0.813 A | 0.939 A |
| $P_{out}$ | 170.24 W | 196.88 W |
| Convergence time | 0.05 s | 0.1 s |

## 4. Conclusion

In this paper, the objective was to use the principle of current controlled voltage source and the parallel multicellular converter to improve the energy efficiency of a photovoltaic chain. A method by modification of the technics based on fraction of the short circuit current (FSCC) with the aim of generating a reference current to control the inductance current in order to control the static converter to follow the MPP has been proposed, this reference current was used as reference current in sliding mode control. The parallel multicellular converter integrating synchronous rectification through the advantages offered by the interleaving mode has

been used to reduce power oscillations and switching losses in switches. The simulation results obtained show the effectiveness of the method by estimating the photocurrent as a reference current used for the pursuit of MPP, responds effectively whatever the environmental conditions, and the contribution of the multicellular converter in reducing oscillations of power around the MPP in steady state and the losses in the switches. The results show a response time of 0.04 s, power oscillations at maximum point around 0.05 W and efficiency of 99.08% facing the high-gain quadratic boost converter. Then by comparing with the interleaved high-gain boost converter the results show a response time of 0.1 s for the transferred power, a very low output voltage ripples of 0.001% and 98.37% as efficiency of the chain. Multicellular converter integrating synchronous rectification can be used in many power system, the control technique based on the control of inductance current is suitable for static converter which the inductance current should be control for its good functioning. The proposed solution can be connected to a grid by reducing the level of the inverter and active filter.

## Supporting information

**S1 Fig. Characteristics (I-V and P-V) of Kyocera Solar KC200GT photovoltaic module at 25˚C under irradiance variations.**
(TIF)

**S2 Fig. Characteristics (I-V and P-V) of Kyocera Solar KC200GT photovoltaic module at $1000 W/m^2$ under temperature variations.**
(TIF)

**S3 Fig. Magnetics coupler behavior for cells connected in parallel.** (a) one cell. (b) 2 cells. (c) 4 cells. (d) 7 cells.
(TIF)

**S1 Table. Characteristics of Kyocera Solar KC200GT photovoltaic module at 25˚C.**
(DOCX)

**S2 Table. Characteristics of Kyocera Solar KC200GT photovoltaic module at $1000 W/m^2$.**
(DOCX)

## Author Contributions

**Conceptualization:** Geoffroy Byanpambé.

**Data curation:** Geoffroy Byanpambé.

**Formal analysis:** Golam Guidkaya.

**Investigation:** Alexis Paldou Yaya.

**Methodology:** Geoffroy Byanpambé.

**Project administration:** Philippe Djondiné.

**Resources:** Golam Guidkaya.

**Software:** Geoffroy Byanpambé, Mohammed F. Elnaggar.

**Supervision:** Noel Djongyang.

**Validation:** Philippe Djondiné.

**Visualization:** Mohammed F. Elnaggar, Alexis Paldou Yaya.

**Writing – original draft:** Emmanuel Tchindebé.

**Writing – review & editing:** Kitmo.

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
