## [Decision Letter · Decision Letter 0]

7 May 2024

PONE-D-24-09884Use of multicellular converter in PV chain for improving of energy efficiency and minimization of power oscillationsPLOS ONE

Dear Dr. BYANPAMBE,

Thank you for submitting your manuscript to PLOS ONE. After careful consideration, we feel that it has merit but does not fully meet PLOS ONE’s publication criteria as it currently stands. Therefore, we invite you to submit a revised version of the manuscript that addresses the points raised during the review process.

The reviewers have highlighted significant questions and comments regarding your manuscript that require careful consideration. I encourage you to revise your manuscript accordingly. When submitting your revised version, please include a separate file containing your detailed responses to the reviewers' comments, addressing them point by point. 

We look forward to receiving your revised manuscript.

Kind regards,

Mohit Bajaj

Academic Editor

PLOS ONE

“This project is supported via funding from Prince Sattam Bin Abdulaziz University, project

number (PSAU/2024/R/1445).”

Reviewers' comments:

Reviewer's Responses to Questions

**Comments to the Author**

1. Is the manuscript technically sound, and do the data support the conclusions?

Reviewer #1: Partly

Reviewer #2: Yes

Reviewer #3: Partly

Reviewer #4: Partly

Reviewer #5: Yes

Reviewer #6: Yes

Reviewer #7: Yes

2. Has the statistical analysis been performed appropriately and rigorously? 

Reviewer #1: No

Reviewer #2: Yes

Reviewer #3: Yes

Reviewer #4: Yes

Reviewer #5: Yes

Reviewer #6: Yes

Reviewer #7: Yes

3. Have the authors made all data underlying the findings in their manuscript fully available?

Reviewer #1: No

Reviewer #2: Yes

Reviewer #3: Yes

Reviewer #4: Yes

Reviewer #5: Yes

Reviewer #6: Yes

Reviewer #7: Yes

4. Is the manuscript presented in an intelligible fashion and written in standard English?

Reviewer #1: Yes

Reviewer #2: Yes

Reviewer #3: Yes

Reviewer #4: Yes

Reviewer #5: Yes

Reviewer #6: Yes

Reviewer #7: No

5. Review Comments to the Author

Reviewer #1: How does the proposed multicellular parallel boost converter compare to existing solutions in terms of efficiency and cost?

Given the environmental conditions' impact on photovoltaic (PV) generator performance, how does the proposed solution ensure consistent energy output across varying conditions?

The introduction mentions several maximum power point tracking (MPPT) techniques. Can you detail how the proposed solution improves upon these or integrates with them?

Can you elaborate on the specific challenges the multicellular converter addresses in non-linear load applications, such as harmonic currents and chaotic behavior?

How does the introduction of multicellular converters impact the overall lifecycle and maintenance costs of PV systems?

How does the modeling of the multicellular boost converter account for real-world inefficiencies, such as component losses and non-ideal behaviors?

The method section discusses the control of inductance current. Can you explain the advantages of this approach over voltage-based control methods in PV systems?

How does the proposed method ensure stability and robustness in the face of sudden changes in environmental conditions?

Can you discuss the scalability of the presented method? How does it perform when scaled up for larger PV installations?

Given the detailed mathematical modeling, how accessible is this method for practical implementation by engineers in the field?

The results show improved efficiency with multicellular converters. Can you quantify the impact on overall system cost and return on investment?

How do the simulation results compare with real-world testing and validations? Are there any discrepancies, and how were they addressed?

The paper mentions rapid pursuit of the MPP despite irradiance variations. How does this rapid tracking affect the system's longevity and reliability?

Can you detail any specific challenges encountered during the simulation, particularly with modeling environmental variations?

How does the efficiency and performance of the proposed system compare to leading commercial systems currently available on the market?

The conclusion mentions the effectiveness of the method under various environmental conditions. Can you provide more details on limitations or conditions where the proposed solution might not perform optimally?

Given the conclusion's claims about reducing power oscillations and switching losses, what are the anticipated impacts on the broader adoption of PV systems?

Can you elaborate on potential future research directions suggested by the findings of this paper?

How do the authors envision the integration of their solution with existing PV infrastructure, particularly in urban settings?

The conclusion suggests improved energy efficiency. Can you discuss any potential environmental impacts, positive or negative, that might arise from the widespread adoption of this technology?

Reviewer #2: This article discusses the role of multicellular converters in enhancing the quality of electrical energy in photovoltaic systems, aiming to maximize power output with minimal oscillations around the maximum power point and reduced switching losses. Through modeling and simulation in Matlab/Simulink, the proposed solutions demonstrate effective performance under irradiance and temperature fluctuations. The subject matter is intriguing, yet there are minor suggestions for improvement. The specific points to address are outlined below:

Clearly articulate the novelty of the proposed control algorithm in the abstract. It remains ambiguous which aspect of the design is innovative.

The derivation of the control design lacks sufficient explanation. Clarification is needed regarding how the control law was derived, including the steps involved.

Justification for the selection of controller gains is necessary.

Provide further elucidation on the advantages and enhancements of the proposed method and technology. Additionally, compare these findings with existing literature. Acknowledge that the control design techniques employed here are akin to those in other studies, but highlight and discuss the challenges encountered in this research to demonstrate that it is not merely an incremental extension of existing methodologies.

I understand that hardware realization may not be possible due to lack of hardware resources. However, please include a critical discussion on what could be anticipated challenges if the proposed algorithm is realized on a real quadrotor system.

Integrating additional performance indices could offer a more comprehensive verification of the controller's performance.

Enhance the discussion on existing control algorithms in the introduction with recent references, such as those available at https://doi.org/10.1371/journal.pone.0293878, https://doi.org/10.1371/journal.pone.0298093, https://doi.org/10.1109/ACCESS.2023.3344451, https://doi.org/10.3389/fenrg.2023.1293267.

Incorporate a discussion on the limitations of the control law in the conclusions section.

Address the issue of chattering inherent in sliding mode control variants. Reference the ' Maximum Power Extraction from a Standalone Photo Voltaic System via Neuro-adaptive Arbitrary Order Sliding Mode Control Strategy with High Gain Differentiation ' and explore potential mitigation strategies in detail.

Conduct a thorough proofreading of the paper to rectify any typos and enhance linguistic clarity.

Ensure that all abbreviations are defined and explained within the text.

Reviewer #3: My review comments are below:

1. Please consider that the presented boost and interleaved boost converters are conventional converters that cannot transfer high powers and work under limited values of power. The presented parallel cell type application of the boot converter is not a novel idea and has been pesented aleady in detail. Therefore I suggest the authors to focus on the control process and reflex this topic on the title of the paper.

2. The calculation of the efficiency for the proposed converter should be presented in detail. By considering the paper at https://www.sciencedirect.com/science/article/pii/S0142061520313958, authors should present the converter efficiency under different working conditions. Compare your results with this paper.

3. The state of the active and passive components under the proposed controller system should be presented in detail. See the paper at https://link.springer.com/article/10.1007/s00202-024-02295-x, and draw your voltage and current graphics according to figures 4 and 6 of this paper. Compare your results with this apper.

4. The voltage and currents under different duty ratios and dynamic loads should be considered and presented by Matlab Simulink.

5. A table should be presented, and advantages and disadvantages of the proposed controller compared with other conventional controllers. Different parameters like the complexity of the control process in practice, cost of the system, efficiency, etc., can be compared.

6. Present several graphics based on the table in comment 5 and present your results. Readers can understand better the advantages of the suggested controller graphically.

Reviewer #4: This article proposes a method to improve the efficiency of photovoltaic (PV) systems by controlling the current through the inductance of a boost converter. Here's a breakdown of the key points and some suggestions for improvement:

1-While the combination of current control and multicellular converters might be interesting, the novelty of the approach compared to existing methods isn't clearly highlighted.

2-The specific algorithm used for Particle Swarm Optimization (PSO) is not mentioned.

3-The switching frequency isn't explicitly stated, which can impact efficiency.

4-Compare the proposed method with existing MPPT (Maximum Power Point Tracking) techniques in terms of efficiency, response time, and complexity.

5-Discuss the limitations of the proposed method and potential areas for future work.

6- Specify the PSO algorithm used and its parameters.

7- Mention the switching frequency used in the simulations.

8- Consider including simulations with more realistic scenarios like partial shading.

Reviewer #5: The paper is readability.

- the all figures should be high quality.

- to cite the reference such as "described by [[44]-[45]]", please check there are double []!.

- check the journal format

- Fig. 17-21 should be explained in more details.

Reviewer #6: Comments to the Author:

The manuscript is well presented with use of multicellular converter in PV chain for improving efficiency and to reduce power oscillations. The paper can be accepted with minor revision and grammatical corrections. The following points can be followed to update/strengthen the manuscript:

1. The manuscript has some grammatical mistakes - Please check it.

2. Literature review is shallow

3. Authors mentioned thaat the efficiency is 99.65% for Inductance current and 99.6% for sliding method which is not convincing due to the passive components involved in the boost converter.

4. The analysis of traditional DC/DC converters can be presented with respect to various duty cycle with the suggested MPP.

5. Novelty of the proposed work should be established by comparing the same with comparable work. Authors can justify or compare the following MPPT based boost converter and justify how the presented MPP technique is superior to other MPPT approaches.

i. Nagaraja Rao, S., Anisetty, S. K., Manjunatha, B. M., Kiran Kumar, B. M., Praveen Kumar, V., & Pranupa, S. (2022). Interleaved high-gain boost converter powered by solar energy using hybrid-based MPP tracking technique. Clean Energy, 6(3), 460-475.

ii. Veerabhadra, & Nagaraja Rao, S. (2022). Assessment of high-gain quadratic boost converter with hybrid-based maximum power point tracking technique for solar photovoltaic systems. Clean Energy, 6(4), 632-645.

6. In the abstract and conclusion, the results performances should be reflected which helps to improve the quality of the manuscript. Add the results values in the abstract.

Reviewer #7: The authors presented “Use of multicellular converter in PV chain for improving energy efficiency and minimization of power oscillations,” which is very interesting. However, it requires a few suggestions to improve. The suggestions and comments are as follows:

1. The presented work is very good. The title of the paper needs to reflect the work that is presented.

2. The abstract needs to highlight some significant results. It should include 1-2 lines of gaps, authors' contributions, and advantages of the work.

3. The introduction section needs to be improved. Try to avoid bulk referencing (ex: ) [[1]-[4]]) , also, the references should be in chronological order.

4. The literature survey can be improved by adding more related papers. It is advised to add the following papers to improve the introduction section.

https://doi.org/10.1002/oca.2773, https://doi.org/10.1007/s12667-021-00465-5, https://doi.org/10.1007/s40435-023-01274-7,

5. The authors should add more results to show the speed of tracking and the accuracy of tracking.

6. The simulation results should be shown in good-quality figures.

7. In simulation results, to highlight the effectiveness of the proposed MPPT, the authors are recommended to compare their method with other methods that also target fast MPPT convergence to provide a fair comparison. This can highlight the paper's contribution to already existing fast methods for MPT.

8. Detailed circuit parameters should be included in the manuscript.

9. The paper requires further English revision as it has many grammatical mistakes.

6. PLOS authors have the option to publish the peer review history of their article (what does this mean?). If published, this will include your full peer review and any attached files.

Reviewer #1: No

Reviewer #2: **Yes: **Dr. Safeer Ullah

Reviewer #3: No

Reviewer #4: No

Reviewer #5: No

Reviewer #6: **Yes: **Dr. S. Nagaraja Rao

Reviewer #7: No

---

## [Author Response · Author response to Decision Letter 0]

17 Jun 2024

Response to Reviewers comments

Reviewer #1: 

How does the proposed multicellular parallel boost converter compare to existing solutions in terms of efficiency and cost? 

In terms of efficiency, the multicellular parallel boost converter offers a better reduction in conduction losses in switching switches and oscillations reduction at steady state (Fig 2 page 4).

The cost is slightly higher than the single cell boost converter.

Given the environmental conditions' impact on photovoltaic (PV) generator performance, how does the proposed solution ensure consistent energy output across varying conditions?

The reference current is automatically generated by a light sensor. Thus, depending on the level of illumination detected, a reference current value will be provided; Through to the PI regulator, the inductor current will be controlled to follow this value. Depending on the current value, the corresponding voltage value will be imposed. (Fig 19 page 19, Figs 21 and 22 page 21).

The introduction mentions several maximum power point tracking (MPPT) techniques. Can you detail how the proposed solution improves upon these or integrates with them?

The proposed solution is a direct improvement of the short circuit method technique where the current at the power point was given by Eq 21 (page 10):

I_OPT=K_I I_SC (21)

Where the proportional constant K_I depends on the PV cell technology, meteorological conditions

and the fill factor, mainly. However, in many cases, K_I is determined by performing a PV scanning every several minutes. After K_I is obtained, the system remains with the approximation of Eq 21, until the next calculation of K_I.

Eq 21 has been modified taking into account certain parameters to obtain the reference current given by Eq 24 (page 11):

I_ref= I_(ph,ref)=K_G I_OPT (24) 

Où K_G=G/G_r 

The coefficient K_G depends only on the illumination, unlike

K_I which depends on the PV cell technology, meteorological conditions and the fill factor, mainly.

The solution presented offers speed and precision in the search for the maximum point, a reduction in oscillations around the maximum power point.

Ease of implementation, automatic update depending on irradiance

Can you elaborate on the specific challenges the multicellular converter addresses in non-linear load applications, such as harmonic currents and chaotic behavior?

The multicellular converter, through its rapid dynamic response, improves the spectral content, hence reducing ripples (Fig 2). Through to the magnetic coupler used, the harmonic currents are attenuated depending on the number of cells used; because the magnetic coupler considerably reduces the currents therefore the harmonic content is not multiple of the number of cells, in other words the harmonic currents not multiple of the number of cells put in parallel will cross a resistance of very high value unlike the harmonic current multiple of the number of cells in parallel (Fig 5 page 7). According to the authors [56,57] chaotic phenomena were more observed with low switching frequency values. Concerning chaotic phenomena, we have not yet carried out the chaotic analysis with our converter, we plan to do so in perspective.

How does the introduction of multicellular converters impact the overall lifecycle and maintenance costs of PV systems?

Multicellular converters provide better current regulation, enable low current operation in power switches, at low switching frequency. These advantages reduce the heating and oscillation that were once responsible for the destruction of semiconductors. Good energy quality will ensure a good lifespan of the system.

How does the modeling of the multicellular boost converter account for real-world inefficiencies, such as component losses and non-ideal behaviors?

Certain aspects have been neglected in order to facilitate modeling.

The method section discusses the control of inductance current. Can you explain the advantages of this approach over voltage-based control methods in PV systems?

This method uses a reference current generated by the illumination to control the inductor current. Its advantages are the rapid pursuit of the maximum power point and the precision in the search for the maximum power point during rapid changes in conditions climatic. It is easily adaptable compared to other techniques.

How does the proposed method ensure stability and robustness in the face of sudden changes in environmental conditions?

Stability is ensured by a low presence of oscillations around the maximum power point in steady state, robustness for its part is translated by a rapid search for the power point during change. (Fig 19 page 19, Figs 21 and 22 page 21, Fig 26 page 23, Fig 30 page 25)

Can you discuss the scalability of the presented method? How does it perform when scaled up for larger PV installations?

The method works well in a large installation, just set the maximum current value and the reference current is generated automatically. Which makes the approach integrable into any type of photovoltaic system.

Given the detailed mathematical modeling, how accessible is this method for practical implementation by engineers in the field?

To the extent that the inductor current can be measured through a current sensor, the entire control approach can be integrated into a microcontroller in order to control the photovoltaic system in real time. This method is therefore accessible for practical implementation.

The results show improved efficiency with multicellular converters. Can you quantify the impact on overall system cost and return on investment?

Our control approach with a multicellular converter is in line with energy efficiency, allowing it provides to the load the power necessary for its functioning. Although the overall cost of the system will be relatively high, the efficiency obtained can already guarantee the return on investment.

How do the simulation results compare with real-world testing and validations? Are there any discrepancies, and how were they addressed?

We have not yet carried out an experimental validation test on our approach. As we mentioned at the conclusion of our manuscript, we plan to carry out an experimental study of our approach.

The paper mentions rapid pursuit of the MPP despite irradiance variations. How does this rapid tracking affect the system's longevity and reliability?

The rapid pursuit of the MPP in the face of variations in irradiation allows the system to exploit the maximum energy produced by the photovoltaic generator. The performance is therefore improved. Furthermore, the reduction of switching losses in the switches, which will improve the overall lifespan of the system.

Can you detail any specific challenges encountered during the simulation, particularly with modeling environmental variations?

During the simulation a signal generator was used to generate environmental variations. With this approach, the difficulties initially encountered regarding the convergence of the entire PV system were wiped out.

How does the efficiency and performance of the proposed system compare to leading commercial systems currently available on the market?

The proposed approach presents a very high robustness to the variation of environmental conditions and system parameters. Compared to the approaches available on the market, the proposed approach presented presents a better performance.

The conclusion mentions the effectiveness of the method under various environmental conditions. Can you provide more details on limitations or conditions where the proposed solution might not perform optimally?

One of the hypotheses for developing the method was based on the weak influence of temperature on the current; faced with a strong temperature variation, the current continues its reference value but the voltage is very sensitive due to the strong influence of the temperature on the voltage.

Given the conclusion's claims about reducing power oscillations and switching losses, what are the anticipated impacts on the broader adoption of PV systems?

The expected impacts are the reduction in the complexity of the filtering system, the increase in the lifespan of the components, the improvement in the quality of the energy produced, and the reduction in maintenance costs.

Can you elaborate on potential future research directions suggested by the findings of this paper?

Potential future research directions that may arise are:

- Study of an autonomous photovoltaic system;

- Photovoltaic systems with multicellular converter connected to the grid;

- Heating systems.

How do the authors envision the integration of their solution with existing PV infrastructure, particularly in urban settings?

The solution presented will be used as an interface between the photovoltaic generator and the load.

The conclusion suggests improved energy efficiency. Can you discuss any potential environmental impacts, positive or negative, that might arise from the widespread adoption of this technology?

Improving energy quality will help reduce noise caused by harmonics and reduce the number of fires resulting from overheating in electrical systems. 

Thank you Sir for your comments and suggestions.

Reviewer #2: 

This article discusses the role of multicellular converters in enhancing the quality of electrical energy in photovoltaic systems, aiming to maximize power output with minimal oscillations around the maximum power point and reduced switching losses. Through modeling and simulation in Matlab/Simulink, the proposed solutions demonstrate effective performance under irradiance and temperature fluctuations. The subject matter is intriguing, yet there are minor suggestions for improvement. The specific points to address are outlined below:

Clearly articulate the novelty of the proposed control algorithm in the abstract. It remains ambiguous which aspect of the design is innovative.

The derivation of the control design lacks sufficient explanation. Clarification is needed regarding how the control law was derived, including the steps involved.

Justification for the selection of controller gains is necessary.

Provide further elucidation on the advantages and enhancements of the proposed method and technology. Additionally, compare these findings with existing literature. Acknowledge that the control design techniques employed here are akin to those in other studies, but highlight and discuss the challenges encountered in this research to demonstrate that it is not merely an incremental extension of existing methodologies.

I understand that hardware realization may not be possible due to lack of hardware resources. However, please include a critical discussion on what could be anticipated challenges if the proposed algorithm is realized on a real quadrotor system.

Integrating additional performance indices could offer a more comprehensive verification of the controller's performance.

Enhance the discussion on existing control algorithms in the introduction with recent references, such as those available at https://doi.org/10.1371/journal.pone.0293878, https://doi.org/10.1371/journal.pone.0298093, https://doi.org/10.1109/ACCESS.2023.3344451, https://doi.org/10.3389/fenrg.2023.1293267.

Incorporate a discussion on the limitations of the control law in the conclusions section.

Address the issue of chattering inherent in sliding mode control variants. Reference the ' Maximum Power Extraction from a Standalone Photo Voltaic System via Neuro-adaptive Arbitrary Order Sliding Mode Control Strategy with High Gain Differentiation ' and explore potential mitigation strategies in detail.

Conduct a thorough proofreading of the paper to rectify any typos and enhance linguistic clarity.

Ensure that all abbreviations are defined and explained within the text.

Thank you Sir for these comments and suggestions, we have improved the manuscript by making many modifications, while also taking your suggestions into account. The references added in the introduction are references [38-41]

Reviewer #3: 

My review comments are below:

1. Please consider that the presented boost and interleaved boost converters are conventional converters that cannot transfer high powers and work under limited values of power. The presented parallel cell type application of the boot converter is not a novel idea and has been pesented aleady in detail. Therefore I suggest the authors to focus on the control process and reflex this topic on the title of the paper.

Thank you Sir for your clarification, your suggestion has been taken into account and integrated into the manuscript.

2. The calculation of the efficiency for the proposed converter should be presented in detail. By considering the paper at https://www.sciencedirect.com/science/article/pii/S0142061520313958, authors should present the converter efficiency under different working conditions. Compare your results with this paper.

The calculation of the efficiency is presented on pages 14-15 of the manuscript, the efficiency of the converter in different working conditions according to reference [49] are presented on page 19 (Fig 19), page 21 (Figs 21 and 22).

3. The state of the active and passive components under the proposed controller system should be presented in detail. See the paper at https://link.springer.com/article/10.1007/s00202-024-02295-x, and draw your voltage and current graphics according to figures 4 and 6 of this paper. Compare your results with this paper.

Thank you Sir for your clarification, your suggestion has been taken into account and integrated into the manuscript.

4. The voltage and currents under different duty ratios and dynamic loads should be considered and presented by Matlab Simulink.

The voltage and currents under different duty cycles are shown in the figures (Figs 10-12) on pages 16-17. Thank you for this remark.

5. A table should be presented, and advantages and disadvantages of the proposed controller compared with other conventional controllers. Different parameters like the complexity of the control process in practice, cost of the system, efficiency, etc., can be compared.

Thank you for this remark. The comparisons are presented in tables 3 and 4 of the manuscript on page 20.

6. Present several graphics based on the table in comment 5 and present your results. Readers can understand better the advantages of the suggested controller graphically.

Thank you for this remark, several graphs have been added to present the results.

Reviewer #4: 

This article proposes a method to improve the efficiency of photovoltaic (PV) systems by controlling the current through the inductance of a boost converter. Here's a breakdown of the key points and some suggestions for improvement:

1-While the combination of current control and multicellular converters might be interesting, the novelty of the approach compared to existing methods isn't clearly highlighted.

2-The specific algorithm used for Particle Swarm Optimization (PSO) is not mentioned.

3-The switching frequency isn't explicitly stated, which can impact efficiency.

4-Compare the proposed method with existing MPPT (Maximum Power Point Tracking) techniques in terms of efficiency, response time, and complexity.

Thank you for your comments, the document has been revised and we have incorporated the shortcomings that you highlighted. The comparisons were made and presented in tables 3 and 4 (page 20).

5-Discuss the limitations of the proposed method and potential areas for future work.

The control technique is more applicable to boost converters and its derivatives, as well as to the converter requiring control of the inductor current for their proper operation.

The proposed method can be used in battery storage systems, for connection to the electrical network.

6- Specify the PSO algorithm used and its parameters.

The PSO algorithm used is presented in Figure 8 as well as its parameters on page 12.

7- Mention the switching freq

---

## [Decision Letter · Decision Letter 1]

13 Aug 2024

A modified fractional short circuit current MPPT and multicellular converter for improving power quality and efficiency in PV chain

PONE-D-24-09884R1

Dear Dr. BYANPAMBE,

We’re pleased to inform you that your manuscript has been judged scientifically suitable for publication and will be formally accepted for publication once it meets all outstanding technical requirements.

Kind regards,

Dr. Mohit Bajaj

Academic Editor

PLOS ONE

Additional Editor Comments (optional):

Reviewers' comments:

Reviewer's Responses to Questions

**Comments to the Author**

1. If the authors have adequately addressed your comments raised in a previous round of review and you feel that this manuscript is now acceptable for publication, you may indicate that here to bypass the “Comments to the Author” section, enter your conflict of interest statement in the “Confidential to Editor” section, and submit your "Accept" recommendation.

Reviewer #2: All comments have been addressed

Reviewer #3: All comments have been addressed

Reviewer #4: All comments have been addressed

Reviewer #5: All comments have been addressed

Reviewer #6: All comments have been addressed

Reviewer #7: All comments have been addressed

2. Is the manuscript technically sound, and do the data support the conclusions?

Reviewer #2: Yes

Reviewer #3: Partly

Reviewer #4: Partly

Reviewer #5: Yes

Reviewer #6: Yes

Reviewer #7: Yes

3. Has the statistical analysis been performed appropriately and rigorously? 

Reviewer #2: Yes

Reviewer #3: Yes

Reviewer #4: N/A

Reviewer #5: Yes

Reviewer #6: Yes

Reviewer #7: Yes

4. Have the authors made all data underlying the findings in their manuscript fully available?

Reviewer #2: Yes

Reviewer #3: Yes

Reviewer #4: (No Response)

Reviewer #5: Yes

Reviewer #6: Yes

Reviewer #7: Yes

5. Is the manuscript presented in an intelligible fashion and written in standard English?

Reviewer #2: Yes

Reviewer #3: Yes

Reviewer #4: (No Response)

Reviewer #5: Yes

Reviewer #6: Yes

Reviewer #7: Yes

6. Review Comments to the Author

Reviewer #2: The authors have made commendable efforts to revise the paper in accordance with the reviewers' comments. However, I was unable to locate their point-by-point responses to each question.

Reviewer #3: While some comments were not addressed and incorporated in the revision, the revised manuscript is in a better state compared to the initially submitted paper and can be considered for publication.

Reviewer #4: I don't have any additional comments for the author including concerns about dual publication, research ethics, or publication ethics.

Reviewer #5: All comments are addressed and the all figures are improved but still the figures can be expanded in order to see clearly. The paper is well revised. It is very interesting.

Reviewer #6: Authors have answered all the queries raised by the reviewers.

I feel that this manuscript is now acceptable for publication.

Reviewer #7: Appreciate all authors for their efforts to address all comments and suggestions.. the revised version of the manuscript is very well in the standards

7. PLOS authors have the option to publish the peer review history of their article (what does this mean?). If published, this will include your full peer review and any attached files.

Reviewer #2: **Yes: **Safeer Ullah

Reviewer #3: No

Reviewer #4: No

Reviewer #5: No

Reviewer #6: **Yes: **Dr. S. Nagaraja Rao

Reviewer #7: No

---

## [Editor Report · Acceptance letter]

23 Aug 2024

PONE-D-24-09884R1 

PLOS ONE

Dear Dr. BYANPAMBE, 

I'm pleased to inform you that your manuscript has been deemed suitable for publication in PLOS ONE. Congratulations! Your manuscript is now being handed over to our production team.

Kind regards, 

on behalf of

Dr. Mohit Bajaj 

Academic Editor

PLOS ONE